# Pore-forming Esx proteins mediate toxin secretion by *Mycobacterium tuberculosis*

Uday Tak[1], Terje Dokland [1] & Michael Niederweis [1✉]

*Mycobacterium tuberculosis* secretes the tuberculosis necrotizing toxin (TNT) to kill host cells. Here, we show that the WXG100 proteins EsxE and EsxF are essential for TNT secretion. EsxE and EsxF form a water-soluble heterodimer (EsxEF) that assembles into oligomers and long filaments, binds to membranes, and forms stable membrane-spanning channels. Electron microscopy of EsxEF reveals mainly pentameric structures with a central pore. Mutations of both WXG motifs and of a GXW motif do not affect dimerization, but abolish pore formation, membrane deformation and TNT secretion. The WXG/GXW mutants are locked in conformations with altered thermostability and solvent exposure, indicating that the WXG/GXW motifs are molecular switches controlling membrane interaction and pore formation. EsxF is accessible on the bacterial cell surface, suggesting that EsxEF form an outer membrane channel for toxin export. Thus, our study reveals a protein secretion mechanism in bacteria that relies on pore formation by small WXG proteins.

[1] Department of Microbiology, University of Alabama at Birmingham, 845 19th Street South, Birmingham, AL 35294, USA. ✉email: mnieder@uab.edu

Almost all pathogenic bacteria secrete protein toxins that modify or inhibit host cell processes to enable bacterial survival and replication[1,2]. A critical step of exotoxin biogenesis is secretion out of the bacterial cell[3]. Gram-negative bacteria often employ dedicated machineries for toxin secretion, such as the type III and type IV secretion systems, while some toxins, such as the AB-type toxins, are secreted by the general secretion pathway[4]. A wealth of studies have characterized these secretion systems in exquisite functional and structural detail[5–7].

We recently described the *Mycobacterium tuberculosis* (Mtb) outer membrane protein CpnT (channel protein with necrosis-inducing toxin), which contains an N-terminal channel domain and a C-terminal toxin domain[8]. CpnT is produced by Mtb in infected macrophages[8]. Its C-terminal domain, the tuberculosis necrotizing toxin (TNT), is secreted into the macrophage cytosol where it hydrolyzes NAD$^+$ (ref. [9]) and subsequently induces necroptosis[10]. TNT is the only known exotoxin of Mtb, and is required by Mtb to survive and replicate in macrophages[8]. However, it is unknown how TNT is secreted across the Mtb cell envelope. The *cpnT* gene is located in an operon downstream of the *esxF* and *esxE* genes (Fig. 1a). The encoded EsxE and EsxF proteins belong to the WXG100 family of small Esx proteins, which include known virulence factors, such as ESAT-6 and CFP-10 (refs. [11,12]). The *esxF* and *esxE* genes are upregulated in latent and active granulomas of Mtb-infected macaques, indicating their importance in the pathogenesis of tuberculosis[13]. However, the molecular function of these "orphaned" *esx* genes is unknown[14]. Since many of the small Esx proteins are associated with the type VII secretion systems of Mtb, also known as ESX systems[15–17], we hypothesized that EsxE and EsxF play a role in toxin secretion by Mtb.

In this study, we discovered that EsxE and EsxF interact with membranes, and form hetero-oligomeric transmembrane pores. Pore formation is required for outer membrane localization of CpnT and toxin secretion by Mtb in infected macrophages. The WXG motifs of both EsxE and EsxF are essential for membrane interaction, and for the channel activity of the EsxE-EsxF complex, thus defining a biochemical role for the previously enigmatic WXG motif. We defined biochemical requirements for pore formation by the EsxE-EsxF complex and identified intermediate states of the transition from water-soluble EsxE-EsxF prepores to a transmembrane pore. These results suggest a dynamic mechanism of pore formation by small Esx proteins that might be applicable to other members of the large WXG100 protein family[18]. Collectively, our results reveal an essential structural role of EsxE-EsxF in CpnT export and toxin secretion by Mtb, and indicate that the EsxE-EsxF complex is part of a mycobacterial protein secretion system.

## Results

**EsxE and EsxF are required for translocation of CpnT to the cell surface of Mtb.** To examine whether EsxE and EsxF play a role in toxin secretion by Mtb, we deleted the entire *cpnT* operon (Fig. 1a) in the strain Mtb H37Rv mc$^2$6206 (Fig. S1 and Table S1). We then reconstituted the individual operon genes in the resulting mc$^2$6206 Δ*esxFE-cpnT-ift* strain (ML2016, Table S2), and examined the effects on CpnT export and TNT secretion. Intracellular CpnT levels were markedly reduced in the absence of either *esxE* or *esxF*, whereas the antitoxin IFT (immunity factor to TNT) changed only marginally (Fig. 1b). EsxE and EsxF were only detected when both genes were expressed together in the bicistronic operon (Fig. 1b), indicating that these proteins likely stabilize each other similar to other Esx protein pairs, such as EsxA and EsxB[19–21]. Collectively, these results show that both EsxE and EsxF are required to maintain CpnT protein levels in Mtb.

Esx proteins have similar four-helix bundle structures as bacterial chaperones[22]. To examine whether the reduced CpnT protein levels in the absence of EsxE and EsxF might have been caused by protein misfolding, we measured the NAD$^+$ glycohydrolase activity of cell lysates after heat treatment at 65 °C to remove the heat-labile antitoxin IFT[9,23]. Enzymatic activity was detected in all samples after heat treatment, indicating that folded CpnT was present even in the absence of EsxE and EsxF (Fig. 1c). To examine whether low CpnT levels in the absence of EsxE and EsxF were caused by impaired export of CpnT, we used flow cytometry to detect surface-exposed TNT, as shown previously[8]. Complementing the Δ*esxFE-cpnT-ift* strain with the entire operon significantly increased TNT levels on the cell surface compared to expression of *cpnT-ift* alone (Fig. 1d). Fluorescence microscopy of the same Mtb strains stained with an α-TNT antibody revealed TNT on the surface of Mtb only in the presence of EsxE and EsxF (Fig. 1e, f). These results indicate that EsxE and EsxF are involved in translocating CpnT to the cell surface of Mtb.

**EsxE and EsxF are required for TNT secretion into the cytosol of macrophages infected with Mtb.** The lack of surface TNT localization in the absence of EsxE-EsxF may also impact TNT secretion during intracellular infection. To test this hypothesis, THP-1 macrophages were infected with Mtb, and TNT secretion was examined by fluorescence microscopy, as reported previously[9]. The plasma membrane of THP-1 macrophages was permeabilized with digitonin 48 h after infection to enable access of antibodies to cytosolic antigens. Cytosolic TNT was only detected in macrophages infected with the Mtb strain expressing the complete *esxF-esxE-cpnT-ifT* operon, but not in macrophages infected with Mtb expressing *cpnT-ift* alone (Fig. 1g, h). This secretion defect was also observed during in vitro growth of Mtb (Fig. S1c). These results established that EsxE and EsxF are required for TNT secretion by Mtb.

**Conserved WXG motifs are essential for the function of EsxE and EsxF.** To understand the mechanism by which EsxE and EsxF mediate CpnT secretion, we next focused on the EsxE and EsxF proteins. The crystal structure of the EsxE-EsxF complex of *Mycobacterium abscessus* is similar to the structures of other Mtb WXG100 proteins, such as ESAT-6 (EsxA) and CFP-10 (EsxB), in which each protein contributes an α-helical hairpin to a four-helix bundle[24]. The WXG100 protein family is characterized by the presence of a WXG (tryptophan-X-glycine) motif and a size of ~100 amino acids (Fig. 2a)[12,25]. The WXG motif has been implicated in dimerization of Esx proteins[22]. EsxF also contains a YXXXD/E motif (Fig. 2a), which is a secretion signal of ESX-1 substrates[26]. To examine the role of these motifs in the function of EsxE and EsxF, we generated mutated bicistronic *esxF-esxE* operons and integrated them at the mycobacteriophage L5 attachment site into an Mtb strain carrying the *cpnT-ift* genes in an episomal vector. Mutation of the WXG motifs of EsxF (*esxF*$_{W48A}$-*esxE*) and EsxE (*esxF-esxE*$_{W38A}$) decreased intracellular CpnT levels, and moderately decreased EsxF and EsxE protein levels in whole-cell lysates (WCL) of Mtb (Fig. 2b, c). In contrast, mutation of the YXXXE motif in EsxF (*esxF*$_{Y86A}$-*esxE*) did not alter CpnT and EsxF protein levels (Fig. 2b, c).

Next, we examined whether these motifs affected TNT secretion. No cytosolic TNT was detected in macrophages infected with Mtb when the WXG motifs of EsxF or EsxE were mutated (Fig. 2d, e). In contrast, mutation of the YXXXE motif in EsxF (*esxF*$_{Y86A}$) did not reduce TNT secretion in macrophages (Fig. 2d, e). These results established that the WXG motifs of EsxE and EsxF are essential for TNT secretion, in contrast to the

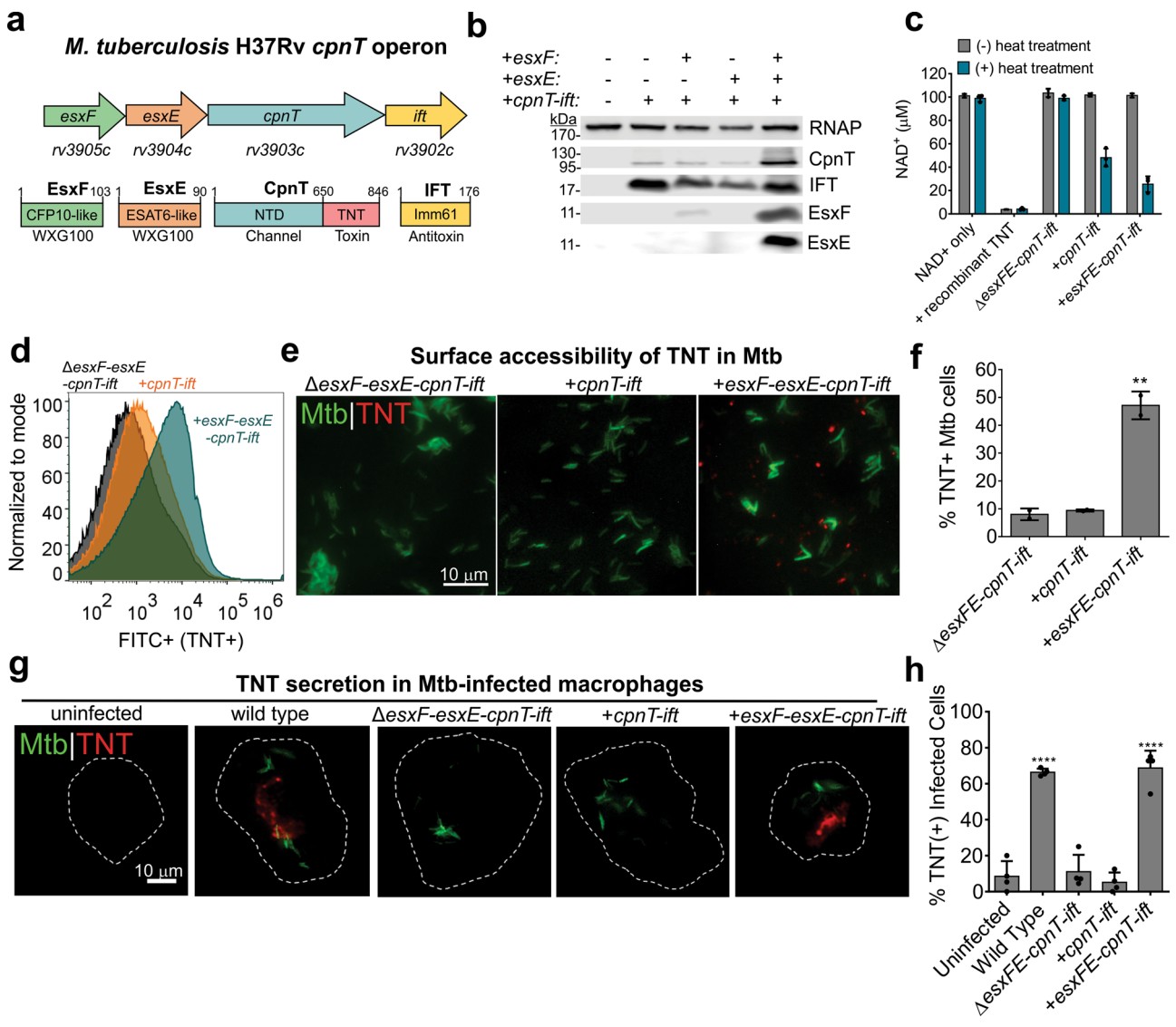

**Fig. 1 EsxE-EsxF are required for TNT surface accessibility and secretion by *M. tuberculosis*. a** The *cpnT* operon of Mtb and the domain organization of the encoded proteins. TNT tuberculosis necrotizing toxin, IFT immunity factor to TNT, NTD N-terminal domain. **b** CpnT protein levels are dependent on EsxE and EsxF. Immunoblot of Mtb whole-cell lysates detected by antibodies specific for the indicated proteins. CpnT was detected using an anti-TNT antibody for the C-terminal domain. RNA polymerase (RNAP) was used as a loading control. Representative of two experiments. **c** The $NAD^+$ glycohydrolase activity of Mtb is dependent on EsxE and EsxF. The $NAD^+$ glycohydrolase activity of whole-cell lysates of Mtb strains was determined without or with heat treatment at 65 °C to release the antitoxin from TNT. The residual $NAD^+$ concentration was measured by conversion of $NAD^+$ to a fluorescent intermediate after NaOH treatment. $NAD^+$ with only buffer and with added recombinant TNT were used as negative and positive controls, respectively. Representative experiment shown from two separate trials. **d, e** Surface accessibility of TNT of the indicated Mtb strains by flow cytometry using an α-TNT antibody and FITC-conjugated secondary antibody (**d**) and by fluorescence microscopy of Mtb strains stained with DMN-trehalose (green), and probed with α-TNT and Alexafluor-594-conjugated secondary antibody (**e**). **f** Quantification of **e**. Percentage of TNT-positive cells out of total bacteria from at least five fields of view. $N \geq 1000$ bacterial cells total over two independent experiments. Shown is mean and standard deviation from two experiments. The statistical analysis was performed using one-way ANOVA analysis with Dunnet's multiple comparison test using the *ΔesxFE-cpnT-ift* strain as the negative control. $P = 0.0018$. **g** Detection of TNT in the cytosol of Mtb-infected macrophages is dependent on EsxE and EsxF. Shown are THP-1 macrophages 48 h after infection with Mtb. THP-1 cells were treated with digitonin to only permeabilize the plasma membrane and were probed with anti-TNT antibody (Alexafluor-594, red). **h** Quantification of **g**. Infected macrophages were scored as TNT+ or TNT− based on the presence or absence of bright red punctae and quantified as % TNT-positive cells. At least 100 cells were analyzed of at least four independent experiments. Data shown is the mean ± SD for four biological replicates. Statistical analysis was performed using one-way ANOVA with Dunnet's multiple comparison test compared to the uninfected sample as the negative control. $P < 0.0001$. Source data are provided in the Source data file.

YXXXE motif. The lack of secreted TNT in Mtb cells producing WXG motif mutants of EsxE or EsxF might result from a failure of CpnT export to the cell surface. Indeed, Mtb strains with EsxE or EsxF containing mutated WXG motifs had significantly reduced TNT levels on the cell surface compared to Mtb with

the wt *cpnT* operon (Fig. 2f, g). Taken together, these results reveal that the WXG motifs of EsxE and EsxF are required for efficient CpnT translocation to the outer membrane of Mtb and subsequent TNT secretion into the cytosol of infected macrophages.

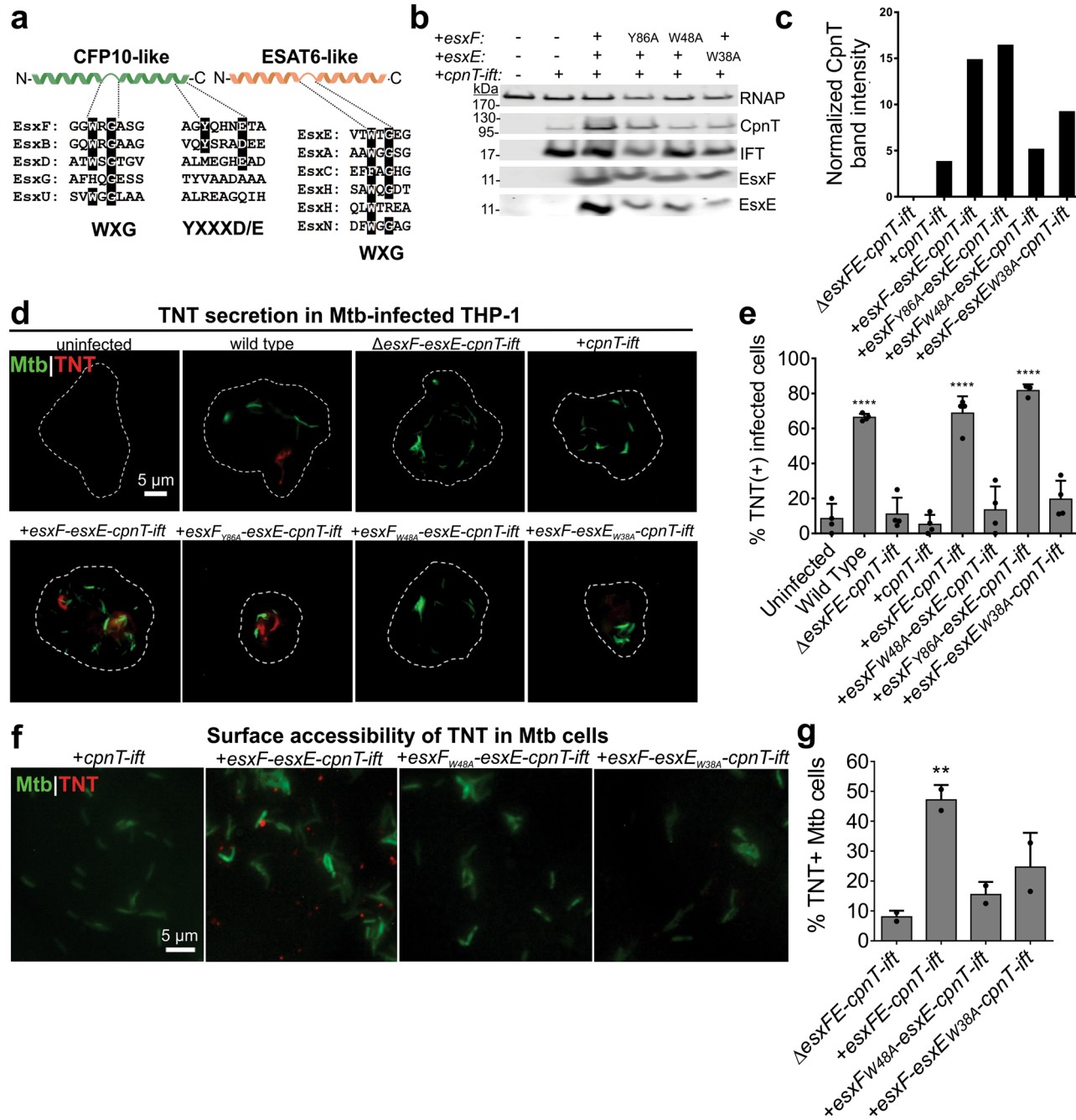

**Fig. 2 The WXG motif is required for EsxE-EsxF function. a** Sequence alignment of the WXG YXXXD/E motifs in Esx heterodimer pairs of *M. tuberculosis*. **b** Immunoblot of Mtb whole-cell lysates probed with antibodies for the indicated proteins. RNA polymerase (RNAP) was used as a loading control. Representative of two experiments **c** Quantification of the CpnT bands from the immunoblot in **b** using ImageJ. **d** Detection of TNT in the cytosol of Mtb-infected macrophages. Representative fluorescence microscopy images of THP-1 macrophages 48 h after infection with DMN-trehalose-labeled Mtb. THP-1 cells were treated with digitonin to only permeabilize the plasma membrane and were probed with anti-TNT antibody (Alexafluor-594, red). Representative of four independent experiments. **e** Quantification of **d**. Infected macrophages were scored as TNT+ or TNT− based on the presence or absence of bright red punctae and quantified as % TNT-positive cells. At least 100 cells were analyzed of at least four independent experiments. Data shown is the mean ± SD from four independent experiments. Statistical analysis was performed using one-way ANOVA with Dunnet's multiple comparison test using the uninfected sample as the negative control. $P < 0.0001$. **f** Surface acccessibility of TNT of the indicated Mtb strains by fluorescence microscopy with DMN-trehalose (green), and probed with α-TNT and an Alexafluor-594-conjugated secondary antibody. Data are from at least ten fields of view from two independent experiments. **g** Quantification of **f**. Percentage of TNT-positive cells out of total bacteria from at least five fields of view. $N \geq$ 1000 bacterial cells total over two independent experiments. Shown is mean and standard deviation from two experiments. The statistical analysis was performed using one-way ANOVA analysis with Dunnet's multiple comparison test using the Δ*esxFE-cpnT-ift* strain as the negative control. $P = 0.016$. Source data are provided in the Source data file.

**EsxE and EsxF are surface accessible proteins and are co-secreted in macrophages.** Our data showed that EsxE and EsxF mediate secretion of CpnT via an unknown mechanism involving the WXG motif. To gain insight into the molecular function of EsxE and EsxF, we investigated the subcellular localization of these proteins in Mtb. As expected, CpnT is localized in the membrane fraction when the *cpnT* and *ift* genes are expressed, as

indicated by the membrane protein MctB and the water-soluble GlpX, which were used as fractionation markers (Fig. 3a). Surprisingly, when the *esxF-esxE* genes are co-expressed with *cpnT* and *ifT* in the native operon, large quantities of CpnT are detectable both in the membrane-bound and water-soluble protein fractions. A possible explanation for the existence of water-soluble CpnT in Mtb are unfolded CpnT polypeptides in transit

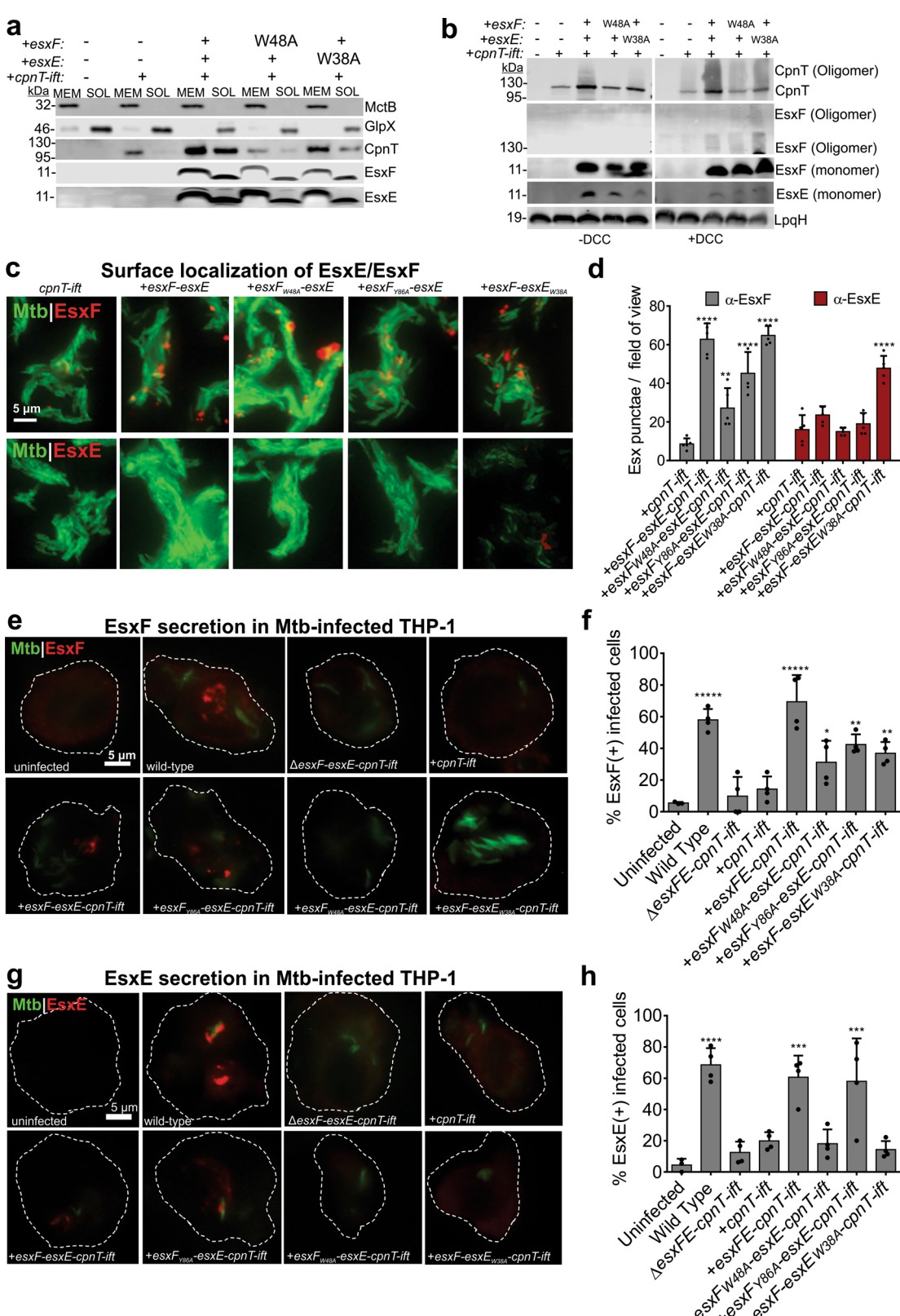

**Fig. 3 Subcellular localization and secretion of EsxE and EsxF. a** Subcellular localization of the indicated Mtb strains. Immunoblot analysis shows water soluble (SOL) and membrane (MEM) fractions obtained by ultracentrifugation. GlpX and MctB were used as water-soluble and membrane-bound controls, respectively. CpnT was detected using an antibody against the TNT domain. Representative blots shown from two independent experiments. **b** Detection of oligomeric CpnT and EsxF in immunoblots of Mtb lysates with or without cross-linking with dicyclohexyl carbodiimide before cell lysis. LpqH served as a loading control. Experiment was performed once. **c** Detection of surface-exposed EsxF or EsxE by fluorescence microscopy of the indicated Mtb strains labeled with the metabolic dye DMN-trehalose, and stained with polyclonal anti-EsxF or EsxE antibody and donkey anti-rabbit-AlexaFluor-594 secondary antibody. Scale = 10 μm. Representative of three experiments shown. **d** Quantification of EsxF or EsxE punctae in images shown in **c** using NIH ImageJ. Data are represented as mean ± SD. *P* < 0.0001 calculated using one-way ANOVA with Dunnet's multiple comparison test as compared to the +*cpnT-ift* strain for nonspecific antibody binding. Representative analysis from one of three separate experiments with identical trends. Detection of secreted EsxF (**e**, **f**) and EsxE (**g**, **h**) in the cytosol of infected macrophages. Fluorescence microscopy of THP-1 macrophages 48 h after infection with DMN-TRE-labeled Mtb. Cells were fixed and treated with digitonin to permeabilize the plasma membrane. Cells were stained with polyclonal anti-EsxF or anti-EsxF antiserum and donkey anti-rabbit-AlexaFluor-594 secondary antibody. The results are representative of four independent experiments. Quantification of the images shown in **f** and **h**. Infected macrophages were scored as EsxF+ and EsxE+ by the presence of bright red punctae compared to the negative controls. The data are represented as % EsxF- and EsxE-positive cells over at least 100 cells from four independent experiments. Data are represented as mean ± SD. *P* < 0.0001 calculated using one-way ANOVA with Dunnet's multiple comparison test. Source data are provided in the Source data file.

to the outer membrane (see also Figs. 1 and 2). Similarly, EsxE and EsxF were also detected in equal amounts in the membrane and water-soluble fractions (Fig. 3a). The distribution of EsxE and EsxF was not affected by mutation of the WXG motifs (Fig. 3a). Cross-linking experiments using dicyclohexyl carbodiimide (DCC) to stabilize protein oligomers revealed that mutation of the WXG motifs of EsxF and EsxE reduced the formation of large CpnT oligomers with apparent molecular weights >110 kDa (Fig. 3b). Fluorescence microscopy using EsxE and EsxF antibodies revealed that EsxF is accessible on the cell surface of Mtb, and that mutation of the WXG or YXXXE motifs did not significantly affect EsxF surface accessibility (Fig. 3c, d). In contrast, wt EsxE was not detectable on the cell surface of Mtb. However, the EsxE$_{W38A}$ mutant was detected on the cell surface of Mtb (Fig. 3c, d). It is possible that the epitopes recognized by the antibodies are somewhat surface-exposed due to conformational changes in the mutated compared to the wt EsxE-EsxF complex.

Next, we examined whether EsxE and EsxF are secreted, like many other small Esx proteins[26–28]. Indeed, we detected secreted EsxF in the cytosol of Mtb-infected macrophages separated from bacteria (Fig. 3e). Disruption of the WXG or the YXXXE motifs reduced secreted EsxF levels compared to the wt and the complemented strains (Fig. 3e, f). This is consistent with our observation that EsxF is surface-exposed independently of the YXXXE motif, although some EsxF molecules may still be released from the surface even with disrupted motifs. In contrast, we detected secreted EsxE only in the presence of the wt *cpnT* operon, but not when the WXG motifs of EsxF or EsxE were mutated (Fig. 3g, h). Taken together these results indicate that EsxE and EsxF form a membrane-associated complex required for CpnT export and are co-secreted. These processes depend, at least partially, on the WXG motifs of EsxF and EsxE.

**The EsxE-EsxF complex forms membrane-spanning pores**. The subcellular localization experiments showed that approximately half of the EsxE and EsxF proteins are membrane associated (Fig. 3a). However, expression of the bicistronic *esxFE* operon in *Escherichia coli* led to the formation of a water-soluble EsxE-EsxF complex that was purified by amylose affinity and ion exchange chromatography (Fig. 4a and Fig. S4a, b). All other known Esx proteins are also water soluble, including EsxE and EsxF from *M. abscessus*[24]. Furthermore, the structure of the *M. abscessus* EsxE-EsxF complex[24] and its derived model for the Mtb EsxE-EsxF heterodimer (Figs. S3b and Fig. 3c) did not show any obvious hydrophobic membrane-spanning domain. However, in silico analysis of the Mtb EsxE-EsxF complex revealed membrane-interacting regions within helix 2 of both EsxF and EsxE and

multiple AXXXA/GXXXG motifs (Fig. S3a), which mediate transmembrane helix packing and oligomerization in α-helical membrane proteins[29,30]. Homology modeling of EsxE-EsxF (Fig. S3b) predicted a membrane-interacting hydrophobic tip comprising the C-terminus of EsxE and the WXG motif of EsxF (Fig. S3a–d). Collectively, these observations suggested that EsxE-EsxF may be able to interact with lipid membranes.

To examine the biochemical properties of EsxE-EsxF, we expressed the bicistronic operon and purified the complex from *E. coli* (Fig. 4a and Fig. S4). Native polyacrylamide gel electrophoresis revealed that the peak fraction after anion exchange chromatography contained predominantly the EsxE-EsxF heterodimer and multiple oligomeric complexes (Fig. 4b). Differential scanning fluorometry confirmed that the proteins were folded at room temperature (Fig. S4c, d). Different oligomeric forms were also evident by gel filtration, upon which EsxE-EsxF eluted at peaks corresponding to molecular weights of ~150 and ~670 kDa (Fig. 4c). Under denaturing conditions, the protein eluted as a single peak at ~44 kDa (Fig. 4c and Fig. S4h). The discrepancies between the apparent molecular weights of the oligomeric forms of EsxE-EsxF observed by native gel compared to gel filtration may be due to the hydrodynamic radius of various EsxE-EsxF oligomers, which may differently impact migration through a polyacrylamide gel compared to a dextran–agarose resin. Electron microscopy and two-dimensional (2D) class averaging analysis of 19,919 negatively stained EsxE-EsxF particles revealed particles of ~10 nm in diameter, with mostly five appendages (Fig. 4d and Fig. S7a) and a central pore of ~3 nm. The higher molecular weight oligomers at ~670 kDa were primarily long filaments together with donut-shaped assemblies (Fig. 4e). These experiments demonstrated that the EsxE-EsxF complex forms an oligomeric pore, and also revealed a propensity to self-assemble into a wide variety of oligomers with varying shapes in vitro. The structural plasticity of EsxE-EsxF is further supported by the almost complete conversion of the purified complex to high molecular weight oligomers after incubation for 2 days at 4 °C (Fig. S4g, h). These oligomers were predominantly long filaments as shown by electron microscopy (Fig. S4i).

The pore-like architecture of the EsxE-EsxF oligomers and the presence of predicted membrane-interacting regions led us to hypothesize that EsxE-EsxF may form channels in lipid membranes. To test this hypothesis we used planar lipid bilayer experiments, as described previously[31]. No channel activity was detected with buffer alone, whereas the water-soluble EsxE-EsxF complex readily formed water-filled transmembrane pores, as observed by a stepwise current increase (Fig. 4f), which is commonly observed for pore-forming toxins[32–34] and porins, such as MspA[35]. In control experiments, we discovered that pore

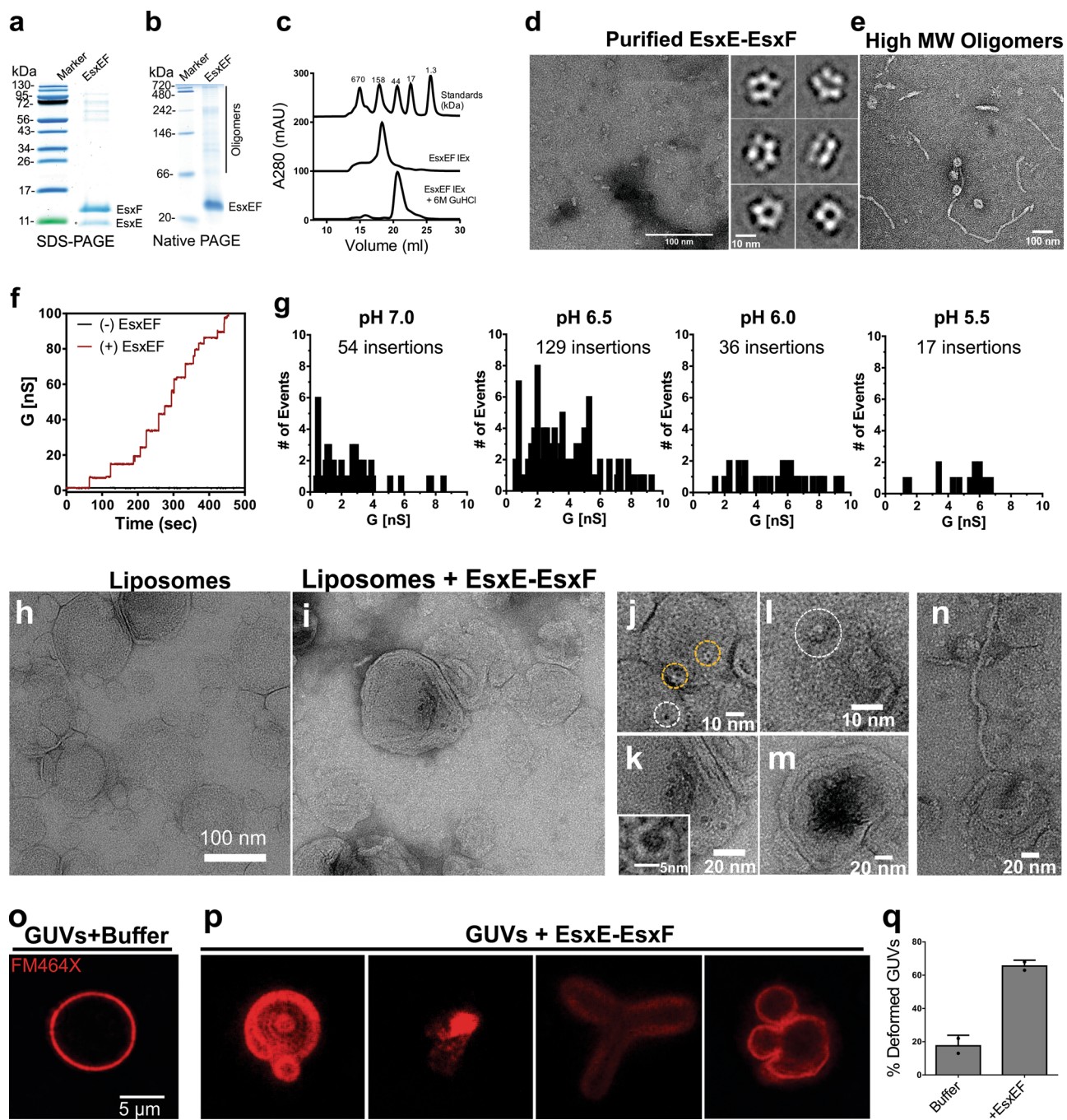

formation by EsxE-EsxF was drastically higher in phosphate buffer compared to HEPES buffer (Fig. S5). Hence, we used phosphate buffer in all subsequent experiments. The channel activity of the EsxE-EsxF complex is strongly pH dependent with a maximal activity at pH 6.5 that is almost tenfold reduced at pH 5.5 (Fig. 4g). These experiments do not distinguish whether the pH dependency of the channel activity of EsxE-EsxF was a result of channel gating or due to a reduced membrane insertion frequency. Differential scanning fluorometry did not detect any changes of the EsxE-EsxF complex at different pH values ranging from pH 5.5–7.0, suggesting that global protein folding is not altered (Fig. S4f). To test whether pH dependence is a consequence of channel gating at lower pH, we incubated purified EsxE-EsxF protein with a lipid bilayer at pH 6.5 until the membrane conductance reached apparent saturation at ~25–30

pores (Fig. S4g). The membrane conductance did not change after lowering the pH to 5.5 by adding HCl (Fig. S4g), suggesting that the EsxE-EsxF pores are not gated at pH 5.5. We concluded that the reduced channel activity of EsxE-EsxF at pH 5.5 is likely caused by pH-induced small conformational changes impairing membrane insertion. Taken together, these experiments demonstrated that the EsxE-EsxF complex forms membrane-spanning pores, whose channel activity is strongly regulated by physiologically relevant environmental conditions, such as pH and phosphate.

**Interactions of the EsxE-EsxF pores with membranes**. To characterize the interactions of the EsxE-EsxF complex with membranes in more detail, we examined EsxE-EsxF pores in liposomes by electron microscopy. Liposomes made from

**Fig. 4 Pore formation by the EsxE-EsxF complex and interaction with membranes. a** Coomassie-stained SDS–polyacrylamide gel and native polyacrylamide gel (**b**) of the purified EsxE-EsxF complex after anion exchange chromatography (representative of more than three purifications). The individual purification steps are shown in Fig. S4. **c** Gel filtration of EsxE-EsxF under native and denaturing conditions indicating oligomerization. **d** Electron microscopy of the EsxE-EsxF complex stained with 1% uranyl formate. Shown are reference-free 2D class averages from 19,919 particles. The full class averages are found in Fig. S7. Data are from a single EM session. Representative of at least three separate experiments. **e** Electron micrograph of the high molecular weight oligomers of EsxE-EsxF (filaments) negatively stained with 1% uranyl acetate. Representative of at least two separate experiments. **f** Current trace of EsxE-EsxF in diphytanoyl phosphatidylcholine bilayers in 25 mM $NaPO_4$ (pH 6.5), 1 M KCl. A total of 1 μg protein was added to both cis and trans compartments, and 10 mV voltage was applied. Results are representative of three protein preparations over ten membranes. **g** Total channel insertion events (membrane pore formation) shown as the number of events as a factor of the conductance values observed at different pH values. Results are representative of at least ten membranes of three independent experiments. The complete profiles with the histograms, channel openings and closings for each condition are shown in Fig. S6. **h, i** Negative stain electron micrographs of dimyristoyl-phosphatidylcholine liposomes with (**i**) or without (**h**) EsxE-EsxF for 30 m at room temperature in 25 mM sodium phosphate (pH 6.5), 150 mM NaCl stained with 1% uranyl formate. The final EsxE-EsxF protein concentration was 50 nM. Representatives of two separate experiments shown. **j–m** Selected events indicating EsxE-EsxF membrane binding or interaction with liposomes. Inset in **k**. Enlarged membrane pore of EsxE-EsxF. **m** Aggregates pores. **n** EsxE-EsxF filaments binding to liposomes. **o** Interaction of EsxE-EsxF with membranes. Giant unilamellar vesicles (GUVs) were treated with the dye FM464X to label the lipid membranes (**o**), and were then incubated with 40 μg EsxE-EsxF (**p**) in a buffer containing 25 mM sodium phosphate (pH 6.5) and 150 mM NaCl. **q** Quantification of the number of deformed GUVs in the presence of EsxE-EsxF. Non-spherical objects which retained the FM464X dye were defined as deformed. The total number of GUVs from two independent experiments was 78 GUVs in buffer and 73 "GUVs" after incubation with EsxE-EsxF. The values are shown as the mean ± standard deviation. Source data are provided in the Source data file.

dimyristoyl-phosphocholine (DMPC) were intact and roughly spherical, as expected (Fig. 4h). In contrast, liposomes incubated with EsxE-EsxF were shriveled, grossly deformed, or destroyed. The few liposomes that appeared to be intact were covered with pore-like features (Fig. 4i), resembling the pentameric pore structure (Fig. 4j) that we observed with the EsxE-EsxF complex in vitro (Fig. 4d). However, the majority of these pores had a larger average diameter of 3.7 ± 1.2 nm and a round particle shape (Fig. 4i, k) compared to the mainly pentameric EsxE-EsxF pores observed in the absence of liposomes (Fig. 4d). We also observed large clusters of pores (Fig. 4m) reflecting the self-assembly propensity of EsxE-EsxF and filaments attached to liposomes (Fig. 4n), suggesting that EsxE-EsxF filaments are able to bind membranes. The electron microscopy experiments support a model in which the water-soluble, mainly pentameric EsxE-EsxF pore interacts with the membrane surface and changes its conformation upon insertion into the bilayer to transition to a transmembrane pore.

To examine membrane destruction by the EsxE-EsxF complex using an alternative approach, we used giant unilamellar vesicles (GUVs) labeled with the dye FM-464X to visualize the membrane. GUVs in a protein-free buffer were unperturbed and spherical, as expected (Fig. 4o), whereas treatment with EsxE-EsxF resulted in a large increase of deformed GUVs (Fig. 4p, q). We observed gross morphological changes, including membrane blebbing, clover-shaped GUVs, and collapsed or ruptured GUVs (Fig. 4p). These results are consistent with our EM observations and reveal that the EsxE-EsxF complex has multiple ways to interact with membranes, including formation of transmembrane pores, membrane remodeling, and/or rupture.

**The WXG motifs are required for pore formation and membrane disruption by the by the EsxE-EsxF complex.** Considering the essential role of the WXG motif in EsxEF-mediated toxin secretion in Mtb, we investigated whether these motifs affected pore formation by the EsxE-EsxF complex. The WXG motifs of EsxF (W48) and EsxE (W38) are located in loops at opposing ends of the structural model of the EsxE-EsxF heterodimer (Fig. 5a). The WXG motif of EsxF is in close contact with the C-terminal domain of EsxE forming a hydrophobic tip (Fig. 5a, b and Fig. S3). Further inspection of this region revealed a GXW motif in EsxE (W89), which is conserved across EsxA paralogs in Mtb (Fig. 5b and Fig. S2). To examine the role of these motifs, we mutated the three associated tryptophan residues. The wt and the

individual mutated heterodimeric EsxE-EsxF complexes were purified from *E. coli* (Fig. S7a, b). Analysis of the mutated EsxE-EsxF complexes in lipid bilayer experiments showed that their pore-forming activity was strongly reduced compared to wt EsxE-EsxF, demonstrating that the WXG and GXW motifs are required for efficient channel formation (Fig. 5c, d and Fig. S6). Interestingly, the few pores observed for EsxF/EsxE$_{W89A}$ complex flickered much more rapidly than the pores of wt EsxE-EsxF, and of the EsxF$_{W48A}$/EsxE and EsxF/EsxE$_{W38A}$ variants (Fig. S6). The opening and closing events of EsxF/EsxE$_{W89A}$ had a much lower channel conductance than those of wt EsxE-EsxF pores (Fig. S6d). These results indicated that the EsxF/EsxE$_{W89A}$ complex can insert into membranes, but is unable to assemble into a stable, mature pore.

To visualize whether the WXG mutations altered the pore structure or the interaction of EsxE-EsxF with membranes, we incubated the purified protein complexes with DMPC liposomes and imaged them by electron microscopy. None of the WXG mutant complexes ruptured or grossly deformed liposomes, in contrast to wt EsxE-EsxF (Fig. 5e–g). However, all WXG mutants complexes appeared to interact with membranes, as observed by the presence of protein particles at the surface of the liposomes (Fig. 5e–g inlets). Many of the EsxE-EsxF$_{W48A}$ and EsxE$_{W38A}$-EsxF complexes appeared to be attached on the surface of the liposomes (Fig. 5e), including filaments formed by EsxE$_{W38A}$-EsxF (Fig. 5f). EsxE$_{W89A}$-EsxF complexes clustered on the liposome surface, where they appeared to form pores of diameters ranging from 1 to 5 nm (Fig. 5g). However, we did not observe any gross deformation of liposomes indicating that these particles likely did not penetrate the membrane, as demonstrated in lipid bilayer experiments (Fig. 5c). The lack of gross membrane disruption was confirmed by treating GUVs with the EsxE-EsxF WXG variants. Fluorescence microscopy did not reveal any major morphological changes in GUVs treated with the WXG variants, as compared to wt EsxE-EsxF, thus indicating a link between pore formation and membrane deformation (Fig. 5h, i). Taken together, these experiments demonstrate that the WXG motifs of EsxF and EsxE are critical for the pore-forming activity and membrane disruption capabilities of the EsxE-EsxF complex.

**The WXG motifs mediate assembly of functional EsxE-EsxF oligomers.** Our data thus far indicated that the WXG motifs of EsxE and EsxF are required for pore-forming activity and,

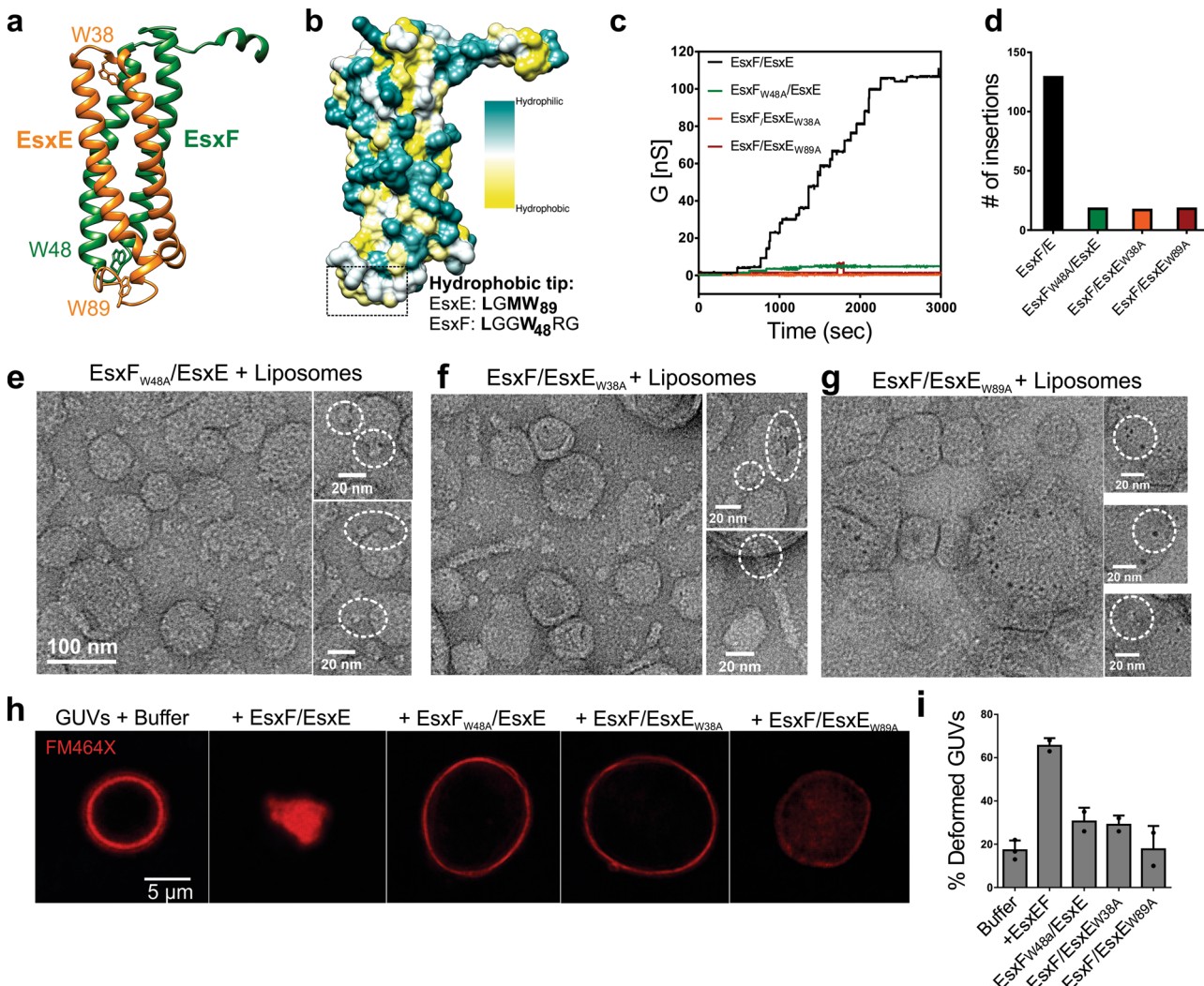

**Fig. 5 The WXG motif is required for pore-forming activity. a** Homology model of EsxE-EsxF indicating the location of the WXG motifs and the EsxE C-terminal GXW motif. The respective tryptophans are highlighted. **b** Hydrophobicity plot of the EsxE-EsxF complex. Some of the residues forming the hydrophobic tip are highlighted. **c** Current trace of wt EsxE-EsxF and the WXG variants in 25 mM sodium phosphate (pH 6.5), 1 M KCl in diphytanoyl phosphatidylcholine bilayers. **d** Total channel insertion events for the indicated proteins. **e–g** Electron micrograph of negatively stained dimyristoyl-phosphatidylcholine liposomes incubated with EsxF$_{W48A}$/EsxE, EsxF/EsxE$_{W38A}$, and EsxF/EsxE$_{W89A}$ at pH 6.5 for 30 min. Selected field of views are shown. Scale: 100 nm unless otherwise indicated. Data are from a single EM session. Representative images are shown from a total of ~25 fields of view per sample. **h** Giant unilamellar vesicles (GUVs) were treated with the dye FM464X to label the lipid membranes and were then incubated with 40 µg wt EsxE-EsxF or the WXG variants in a buffer containing 25 mM sodium phosphate (pH 6.5) and 150 mM NaCl. Representative of two experiments. Representative micrographs are shown. **i** Quantification of the experiments shown in **h**. The values are averages from two independent experiments shown as the mean ± standard deviation. Source data are provided in the Source data file.

subsequently, for CpnT secretion. The shape of the WXG variant particles attached to liposomes (Fig. 5e–g) appeared drastically different from that of wt EsxEF, suggesting an altered conformation which is incapable of forming pores. To gain insight into the conformational dynamics of EsxEF, we generated a pentameric 3D electron density map using single particle analysis resolved to 17.6 Å resolution (Fig. S8a–d). The ultrastructure of the density map clearly resembled a large pore-like object of ~9.5 nm × 6 nm with a central cavity of ~2.5 nm. Rigid-body fitting enabled us to place five copies of the EsxE-EsxF heterodimer unambiguously into the density map (Fig. 6a, b) to generate a model of the EsxEF pore. The organization of the heterodimers suggested that the hydrophobic tip formed by the EsxF WXG motif and the CTD of EsxE form an oligomerization interface between heterodimeric subunits (Fig. 6c). To further examine our model, we generated de novo pentameric EsxE-EsxF complexes

using Rosetta symmetric docking[36]. The lowest free energy computational model also fit the experimental density map and closely resembles the rigid-body fitting model (Fig. S8g, h), in which the WXG motifs are part of a hydrophobic tip forming an oligomeric interface between EsxE and EsxF. This model provided an explanation how the WXG motifs may affect the assembly of EsxE and EsxF into functional pore-forming oligomers. The hydrophobic tip of EsxE-EsxF resembles those found in α-pore-forming proteins, such as VacA, YaxAB, and cytolysin A, which form membrane-spanning oligomers from water-soluble protomers and insert unidirectionally into lipid bilayers (Fig. S9)[37–41]. Importantly, our EM results results show that the EsxE-EsxF complex can exist in many conformations and assemblies ranging from hetero-tetramers, pentamers to large filaments. Thus, the pentameric model represents only one out of several possible EsxE-EsxF conformations.

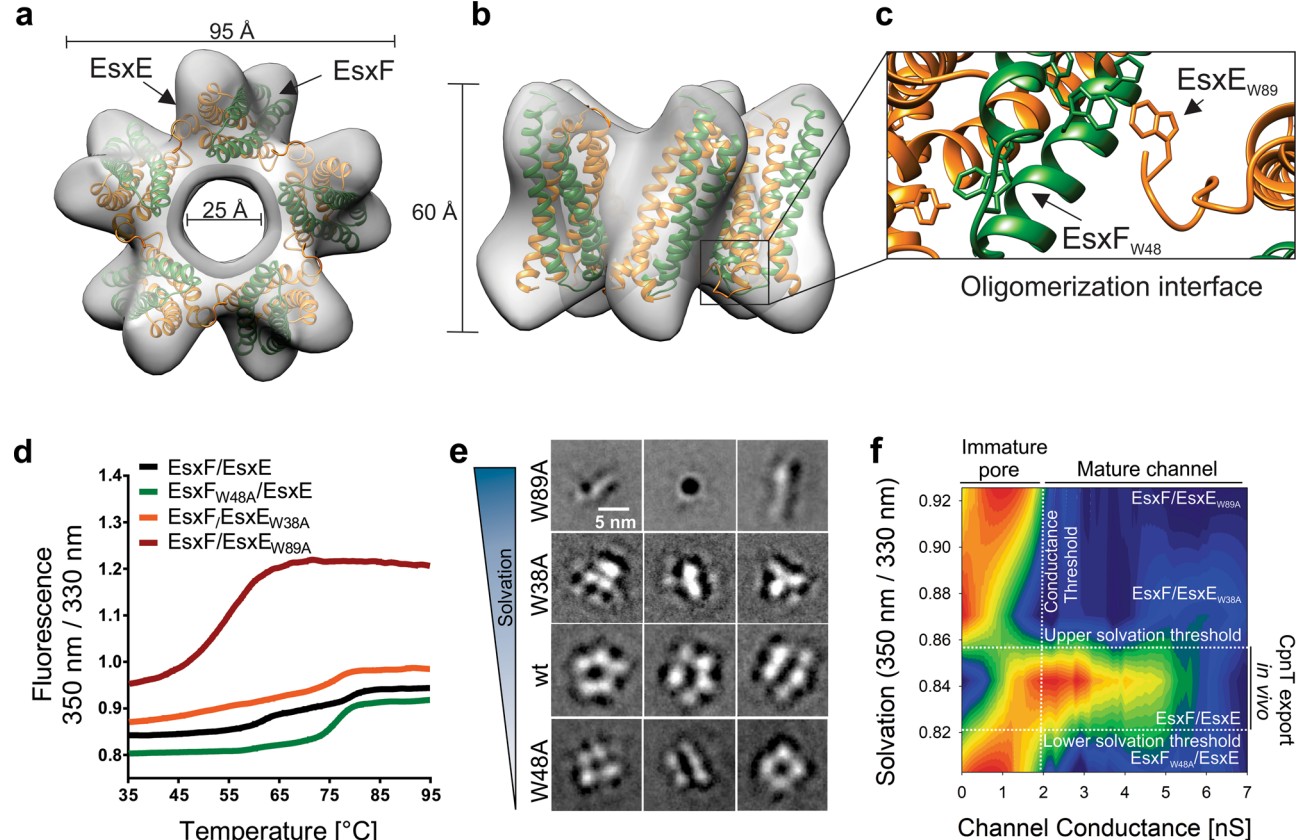

**Fig. 6 Conformational dynamics of EsxE-EsxF pore assembly revealed by electron microscopy. a** Pentameric 3D reconstruction from negatively stained EsxEF particles. The EsxEF homology model was fit to the density using rigid-body fitting in Chimera. Shown is the bottom view. The electron density map for EsxEF is deposited in the electron microscopy data bank (EMDB code: EMD-22727). **b** Side view of the 3D reconstruction shown in **a**. **c** Putative oligomeric interface formed by the WXG motif of EsxF (green) and the CTD of EsxE (orange). **d** Differential scanning fluorimetry of the indicating proteins, indicating the thermostability of the proteins and the water accessibility of the tryptophan residues. **e** Reference-free 2D class averages of the indicated EsxE-EsxF protein particles obtained by negative stain electron microscopy ranked by solvation as determined in **e**. The complete class averages are shown in Fig. S7. Data are from a single EM session. Representative images of at least two experiments are shown. **f** Contour map of the solvation value (350 nm/330 nm) obtained from **d** versus the total number of channel insertions (Fig. S6) for each examined EsxE-EsxF variant. The thresholds were determined based on the event frequency. The WXG variants of EsxE-EsxF form immature pores whereas "mature channel" were formed by wt EsxE-EsxF. The total number of insertions for each protein at each conductance value was smoothed and normalized in Graphpad. Smoothed data were then plotted against the respective solvation values using SigmaPlot. Source data are provided in the Source data file.

To examine whether the WXG motif mutations altered the overall structure of the EsxE-EsxF complex, we used differential scanning fluorometry to measure solvent accessibility of tryptophans as a consequence of protein conformation. The temperature-dependent changes of the intrinsic protein fluorescence revealed that all mutated EsxE and EsxF proteins were folded. However, the water accessibility of the tryptophans ("solvation") in the $EsxE_{W89A}$-EsxF complex was increased and its thermostability was decreased compared to wt EsxE-EsxF, indicating an significantly altered conformation (Fig. 6d). The $EsxF_{W48A}$ mutation had the opposite effect, by decreasing water accessibility to tryptophans and increasing the thermostability of the $EsxE-EsxF_{W48A}$ complex compared to wt EsxE-EsxF (Fig. 6d). The $EsxE_{W38A}$ mutation also increased the thermostability of the $EsxE_{W38A}$-EsxF complex (Fig. 6d). These results indicate that the structures of all mutated EsxE-EsxF complexes are altered in a manner incompatible with pore formation. To examine these conformational changes, we analyzed the structures of the EsxE-EsxF complexes with mutated WXG motifs by electron microscopy. Negatively stained complexes showed oligomers of similar sizes for wt EsxE-EsxF and the $EsxE-EsxF_{W48A}$ mutant (Fig. 6e). However, 2D class averages revealed predominantly four-pronged

particles for the $EsxF_{W48A}$-EsxE complex compared to mainly pentameric wt EsxE-EsxF particles (Fig. 6e). In contrast, the $EsxE_{W89A}$-EsxF complex displayed an elongated cone-like particle (Fig. 6e). This particle contained a widened, cylinder-shaped pore on one end, collapsing into a cone-like shape on the other end. This structure is consistent with the higher solvent accessibility observed by differential scanning fluorometry (Fig. 6d), in which solvent can likely diffuse through the widened pore and reach internal tryptophans. A possible explanation is that the replacement of W89 by the much smaller alanine results in a unidirectional collapse of the hydrophobic domain of the EsxE-EsxF complex (Figs. 5a and 6a). The $EsxE_{W38A}$-EsxF complex formed smaller, asymmetric particles which also stacked into small filamentous tubes (Fig. 6e and Fig. S7). Filament formation is consistent with the observation of high molecular weight oligomers in cross-linking experiments (Fig. 3b). Collectively, these data showed that mutation of all WXG and GXW motifs results in EsxE-EsxF complexes with altered quaternary structures, thus supporting our structural model determined by electron microscopy.

Collectively our biophysical and structural data suggest a mechanism, in which the WXG/GXW motifs form an oligomeric

interface that is required for sequential assembly of an oligomeric complex that is capable of inserting into lipid membranes and of pore formation. Plotting the tryptophan solvation status versus the channel conductance for wt EsxE-EsxF and the WXG mutants revealed that high-conductance channels are only formed at a narrow solvation range occupied by wt EsxE-EsxF prepores (Fig. 6f), whereas channel insertions <2 nS are possible under all solvation conditions, albeit at a much lower frequency. This is supported by our electron microscopy analysis of the WXG mutants which likely represent immature complexes. Pore maturation is characterized by larger channels exceeding conductances of 2 nS and appears to be possible only within a narrow solvation range requiring all three WXG motifs. While wt EsxE-EsxF proteins can access a range of conformations depending on environmental signals, such as pH and/or the presence of small molecules like phosphate, the WXG and GXW mutants assemble into protein complexes that appear to be locked in an immature form. Taken together, these data suggest that channel maturation as observed only for wt EsxE-EsxF is required to export CpnT to the outer membrane, as shown by fluorescence microscopy experiments (Figs. 1 and 2).

## Discussion

**The EsxE-EsxF heterodimer is required for CpnT export in vitro and TNT secretion in vivo.** Our study shows that the EsxE-EsxF heterodimer is essential for CpnT export to the outer membrane and, consequently, for secretion of the toxin TNT. Previously, the small Esx proteins of the WXG100 family were known to be secreted and to exert their functions outside of the Mtb cell. For example, EsxA (ESAT-6) and EsxB (CFP-10) are secreted by their associated type VII secretion system ESX-1 (ref. [17]), and are required for permeabilization of the phagosomal membrane and subsequent escape of Mtb into the cytosol of infected macrophages[42]. Another example are the EsxG and EsxH proteins, which are secreted by the ESX-3 system and are involved in siderophore-mediated iron uptake[43], inhibit protein trafficking by the host endosomal sorting complex required for transport (ESCRT) machinery[28], and block ESCRT recruitment to sites of endolysosomal damage in macrophages infected with Mtb[44]. Here, we show for the first time that small Esx proteins of the WXG100 family have an important molecular function inside the Mtb cell by mediating toxin secretion. The combination of membrane association and surface accessibility indicate that the EsxE-EsxF complex is associated with the outer membrane of Mtb. Proteins are exported across the inner membrane of Mtb by the Sec, TAT, or T7S systems[17]. However, the components of mycobacterial protein secretion systems mediating transport through the periplasm and across the outer membrane are unknown[15,17]. Thus, EsxE and EsxF constitute the first known outer membrane components mediating protein secretion in Mtb.

Considering the association of many small Esx proteins with the T7S systems[16] and the presence of a T7SS signal sequence YXXXD/E[26] in EsxF (Fig. 2a), it is tempting to speculate that the EsxE-EsxF heterodimer might interact with an ESX system to mediate CpnT integration into the outer membrane and TNT secretion. Although mutation of the YXXXD/E motif of EsxF did not alter TNT or EsxF secretion (Fig. 2d, e), it is possible that other motifs target the EsxE-EsxF complex to an as-of-yet unknown ESX system and/or other proteins. Indeed, the C-terminal sequence of EsxF has some similarity to the ESX targeting signal of the ESX-1 substrate EsxB (two out of the seven conserved residues, Fig. S2)[45–47]. It is unclear whether this sequence plays a role in targeting EsxF to an ESX system.

The size of the EsxE-EsxF heterodimer (19 kDa) is small in contrast to other outer membrane proteins mediating protein secretion in Gram-negative bacteria, such as TolC (~53 kDa, T1SS), YscC (67 kDa, T3SS), or the secretins, such as PulD (~70 kDa)[48]. A common feature of these proteins is that they assemble into oligomers. Similarly, the EsxE-EsxF heterodimer assembles to a large variety of small and large complexes, demonstrating a remarkable structural plasticity. The propensity of EsxE-EsxF to form filamentous structures is reminiscent of the ESX-1 substrate EspC, which forms filaments on the cell surface of Mtb[49]. Intriguingly, the presence of filamentous appendages has also been observed in the periplasm of *Mycobacterium smegmatis* by cryo-electron tomography[50]. These findings indicate that filament formation might be a more common property of Esx proteins, although their molecular role has yet to be determined. In the case of EsxE-EsxF, one possibility might be the formation of a channel tunnel, which bridges the inner and outer membrane to export CpnT. It should also be noted that a significant fraction of the EsxE and EsxF proteins are secreted into the cytosol of macrophages infected with Mtb, where it may have biological functions similar to those of other small Esx proteins[15].

**Pore formation by the EsxE-EsxF complex is required to mediate CpnT export and TNT secretion.** Pore formation is a common feature of secretion systems that translocate proteins across bacterial membranes[48]. Subcellular fractionation experiments and fluorescence and electron microscopy with liposomes demonstrate that the EsxE-EsxF complex interacts with lipid membranes. This is consistent with previous observations of the related EsxA-EsxB complex[51,52], indicating that membrane binding is a shared property of small Esx proteins of Mtb. Planar lipid bilayer experiments and electron microscopy experiments of both the purified EsxE-EsxF protein complex alone and associated with liposomes conclusively show that the EsxE-EsxF complex forms pores. Importantly, we found that the channel activity of the EsxE-EsxF complex is very sensitive to environmental conditions, such as pH and buffer composition. For example, the channel activity of the EsxE-EsxF complex was drastically reduced when the pH changed from 6.5 to 5.5 (Fig. 4g) or when phosphate was absent (Fig. S5). Such a sensitivity to environmental conditions may explain why pore formation by EsxA has been a controversial issue for almost two decades. The initial reports indicating that EsxA (ESAT-6) has membrane lytic activity[53,54] were consistent with the critical role of the ESX-1 system in permeabilization of the mycobacterial phagosome[42,55]. Subsequently, pore-forming activity of purified EsxA was observed in liposome efflux experiments[56,57]. However, a recent report concluded that "ESAT-6 does not function as a pore-forming protein or possess intrinsic membrane lytic activity under physiological conditions" and that the previously observed pore-forming activity of EsxA was an artifact of a detergent used during purification[58]. While this conundrum is not yet solved for EsxA, in this study, the EsxE-EsxF complex was purified in the absence of detergents and showed channel activity in lipid bilayer experiments without detergents. Furthermore, we identified point mutations of both EsxE and EsxF, which drastically reduce the channel activity without disrupting complex formation, indicating that distinct biochemical and/or structural properties are required for pore formation by the EsxE-EsxF complex. Point mutations of EsxE and EsxF impairing the channel activity of its complex are also inactive in CpnT translocation and TNT secretion, thus indicating that pore formation by the EsxE-EsxF complex is required for toxin secretion. Electron microscopy revealed a central pore in the EsxE-EsxF complex (Fig. 4d), consistent with the electrophysiology experiments. This pore size of the EsxE-EsxF complex of ~3 nm as derived from the electron microscopy averages is similar to that of TolC-like channels

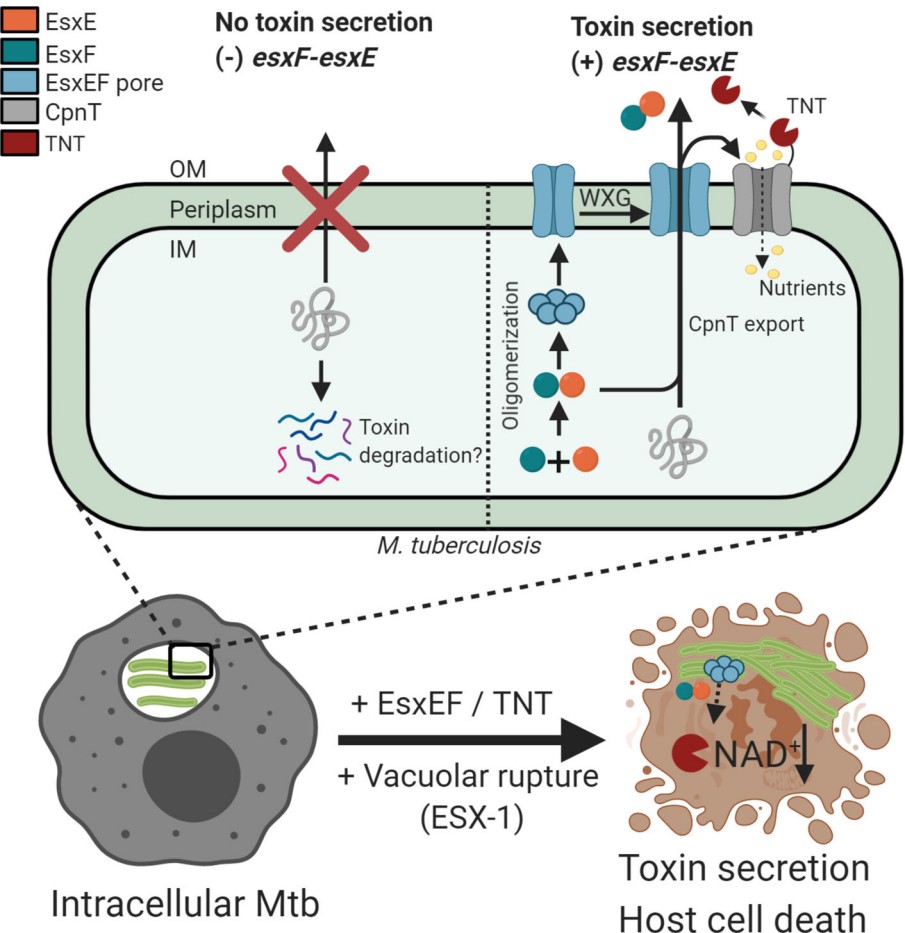

**Fig. 7 Model of EsxE-EsxF mediated toxin secretion by Mtb.** During intracellular infection of macrophages Mtb produces the outer membrane protein CpnT, which enables nutrient uptake and whose secreted C-terminal TNT domain is the major toxin of Mtb. TNT is only toxic when it gains access to the macrophage cytosol after rupture of the phagosomal membrane. EsxE-EsxF heterodimers form large, oligomeric complexes, which bind to membranes and form membrane-spanning pores, which extend to the cell surface. This process is required for export of CpnT to the cell surface of Mtb. EsxEF and CpnT/ TNT are then co-secreted and released into the macrophage cytosol leading to cell death. In the absence of EsxEF, CpnT is mislocalized and is possibly degraded in the cytosol of Mtb to prevent self-intoxication. This figure was created using Biorender.

(3–3.5 nm)[59,60], the outer membrane proteins of type 1 secretion systems[61]. Type 1 secretion systems form a tripartite contiguous passage across two membranes that enables secretion of substrate proteins as unfolded polypeptides[60]. The propensity of the EsxE-EsxF complex to assemble into long filaments (Fig. 4e, j) and to form pores in membranes, indicates that EsxE and EsxF might constitute a channel-tunnel spanning both mycobacterial membranes and the periplasm to enable translocation of CpnT from the cytoplasm of Mtb to the outer membrane. Such a function of EsxE and EsxF would resemble that of TolC-like proteins, which have no homologs in mycobacteria[62]. However, it is unlikely that EsxE and EsxF are sufficient for TNT secretion, since an energy source is required in all known bacterial protein secretion systems[63]. Thus, it is possible that EsxE-EsxF associate with other proteins or protein complexes to achieve CpnT export and TNT secretion.

**The WXG motifs play an essential role in membrane interaction and pore formation by the EsxE-EsxF complex.** Our mutational analysis identified the WXG motifs as essential for pore formation by the EsxE-EsxF complex and for TNT secretion. Electron microscopy of the WXG mutants revealed that mutated WXG motifs alter the structure, thermostability, and solvent

exposure of the EsxE-EsxF complex and its interactions with membranes (Fig. 5e–i). The important role of these motifs for the interaction of EsxE and EsxF with lipid membranes is consistent with many previous observations that tryptophans are critical for interactions of proteins with membranes[64,65]. Alternatively, the WXG motifs could act as molecular switches controlling the conformational changes of the EsxE-EsxF complex depending on environmental signals, such as membrane contact, pH changes, or the presence of phosphate. Remarkably, the mutated EsxE and EsxF proteins are still capable of forming a heterodimer, in contrast to EsxA with a mutated WXG motif, which does not form heterodimers with CFP-10 (ref. [66]). Similarly, mutations of the WXG motifs of YukE, a distant EsxA homolog in *Bacillus subtilis*, abolished its secretion, but did not affect the dimer formation[67].

A model derived from these observations and from the structural analysis of EsxE-EsxF and of the WXG mutants is shown in Fig. 7. During the intracellular infection of macrophages, Mtb exports CpnT to the cell surface. The C-terminal TNT domain of CpnT is secreted into the cytosol of infected macrophages, where its NAD$^+$ glycohydrolase activity kills the host cell[9,10], while the N-terminal CpnT channel enables nutrient uptake. The fact that the amount of IFT protein, encoded by the last gene in the *cpnT* operon, is not affected by the absence of the

*esxF-esxE* genes, indicates that transcription and, due to the likely translational coupling of the overlapping *cpnT-ift* genes[68], also translation of the resulting mRNA are not changed. Thus, in the absence of EsxE and EsxF, CpnT stays in the Mtb cytosol and is either bound by the immunity factor IFT and/or degraded to prevent self-poisoning, as shown previously[9]. In the presence of *esxE* and *esxF*, the EsxE-EsxF heterodimer assembles into an oligomeric prepore in the cytosol of Mtb. We propose that the EsxE-EsxF prepore interacts with the inner membrane where it forms a transmembrane channel, perhaps in complex with an ESX system. This pore enables translocation of further EsxE-EsxF heterodimers into the periplasm which again assemble to a prepore, resulting in pore formation in the outer membrane. Alternatively, the inner membrane channel is extended to span the periplasm via filament formation and connects to EsxE-EsxF pores in the outer membrane exposing EsxF on the cell surface (Fig. 3c). In this model, the putative EsxE-EsxF channel tunnel enables export of the CpnT polypeptide to the outer membrane of Mtb, and subsequent secretion of TNT and EsxE-EsxF. This proposed mechanism resembles type I secretion systems with the outer membrane channel-tunnel TolC bridging the periplasm and connecting to an inner membrane transporter[59,60,69]. Such a mechanism is supported by the observation of membrane-bound EsxE-EsxF filaments (Fig. 4n). It is likely that other proteins associate with EsxE and EsxF. For example, recruitment of an inner membrane transporter is probably necessary to provide energy for the secretion process as has been previously proposed based on the frequent association of distant homologs of EsxA with YukA transporters in *B. subtilis* and other Gram-positive bacteria[11]. Obvious candidates for such a transporter in Mtb are the ESX systems, although functions as chaperones as recently proposed for Esx homologs[22] cannot be excluded for EsxE and EsxF. While it is likely that the EsxE-EsxF complex binds other proteins, the intricate link between channel activity and the function of EsxE-EsxF in CpnT export and TNT secretion suggests a direct functional role for the pore formation.

The WXG mutants of EsxE-EsxF appear to be locked in different stages in the process of formation of the EsxE-EsxF channel tunnel. For example, the EsxE-EsxF$_{W48}$ complex resembles wt oligomers and binds to membranes, but has lost its ability to form ion-conducting channels. The decreased water accessibility of the tryptophans of the EsxF$_{W48A}$/EsxE complex suggests that this variant may be locked in the water-soluble "prepore" state, in which tryptophans are buried in the protein core. Thus, the W$_{48}$XG motif may act as a switch that enables the insertion and/or maturation of the pore in membranes. The EsxE-EsxF$_{W89A}$ complex has an elongated cone-shaped structure with a large and deep cavity, which interacts with membranes, but is unable to form stable transmembrane pores, probably by failing to insert into membranes. By contrast, mutation of the W$_{38}$XG motif decreases the size and symmetry of EsxE-EsxF complexes, and enhances filament formation by the EsxE-EsxF complex.

Our study provides conclusive evidence that the EsxE-EsxF complex forms a membrane channel, which is essential for protein secretion. It is possible that other members of the WXG100 superfamily of bacterial effector proteins[12] have similar properties. Most notably, the previously contested pore formation by EsxA of Mtb[58] is probably functionally important for phagosome membrane permeabilization and Mtb escape from the phagosome[42,55]. Another example is the EsxG-EsxH complex, which is required for iron and zinc uptake by Mtb[43,70], and inhibits antigen export by interfering with ESCRT-mediated endomembrane repair[71]. It is intriguing that the EsxE-EsxF complex is secreted into the macrophage cytosol in addition to its protein secretion function in Mtb. It is possible that the pore-forming activity of the EsxE-EsxF complex aids in the permeabilization of phagosomal, mitochondrial, and plasma membranes to synergize with NAD$^+$ depletion by TNT[10] and to facilitate escape of Mtb from the host cell. In addition to small Esx proteins, the WXG motif is also found in the PE and PPE families of Mtb comprising 168 proteins, the Esp proteins[25] and in Gram-positive bacteria[12]. Thus, our study not only represents a major advancement in our understanding of secretion of TNT and likely of other proteins in Mtb, but also describes a biological function for Esx-paralogs in Mtb and their homologs in the large WXG100 protein family in Gram-positive bacteria.

## Methods

**Plasmid construction.** All cloning procedures were performed using standard restriction enzyme-based methods. PCR products of the indicated genes were amplified using KOD hotstart polymerase using the manufacturer's recommendations and purified using gel extraction (Qiagen). PCR products were digested with PacI/HindIII for mycobacterial expression plasmids and ligated into the appropriate backbone vector (Quickligase, New England Biolabs). For gene expression in *E. coli*, vectors were digested with NdeI/HindIII and ligated. The complete insert including the Shine–Dalgarno sequence were sequenced (UAB Heflin sequencing core). All restriction enzymes and NEB-Quickligase kit were purchased from New England Biolabs. KoD Hotstart Polymerase GC master mix was purchased from Invitrogen. Chemically competent cells (One Shot Mach I) were purchased from Invitrogen and were used for vector cloning. Primers were ordered from IDT Biosciences.

**Strain construction.** Plasmids were transformed into Mtb H37Rv mc$^2$6206 ML2016 (Δ*esxFE-cpnT-IFT*, Tables S1 and S2) via electroporation (500 ng plasmid) and 1 ml of 7H9 + ADS, casamino acids (0.02%), tyloxapol (0.02%), and pantothenate (24 µg/ml)/L-leucine (50 µg/ml; hereafter referred to as AMTB (auxotrophic Mtb) media) was added to the transformation and the culture was grown for 24 h at 500 RPM at 37 °C in a thermoshaker protected from light. Cultures were then plated on AMTB–7H11 agar with the appropriate antibiotic and incubated for 1 month at 37 °C. A single transformant was inoculated in 10 ml of AMTB medium and grown to late log phase, and then seed stocks were made for further experiments and stored at −80 °C in 15% glycerol. Cultures were never passaged more than once. Kanamycin and hygromycin were used at 30 and 50 µg/ml, respectively, for selection.

**Mycobacterial intracellular protein levels and subcellular localization.** Seeds stocks of indicated Mtb strains were inoculated in 30 ml of AMTB growth medium and grown until late log phase (OD = ~1.5) at 37 °C at 200 RPM. The cells were pelleted at 3000 × $g$ and washed twice with PBS (Thermo Fisher Scientific). The cells were then resuspended in 1 ml of 25 mM sodium phosphate 300 mM NaCl pH 7.0 and were bead-beated for two cycles of 30 s at maximum power to induce cell lysis. Samples were incubated on ice in between cycles for 1 m. SDS was added to a 0.1% final concentration and the cells were bead-beated again to solubilize all membrane components. The lysates were then spun at 1000 × $g$ to remove cell debris and the supernatant was considered "WCL". For subcellular localization, SDS was omitted from the buffer and the samples were further ultracentrifuged at 100,000 × $g$ to separate the membrane (pellet) from the soluble fraction (supernatant). The supernatant (soluble) fraction was spun again at 100,000 × $g$ to remove residual membrane contaminants and the supernatant was collected and referred to as soluble fraction. The membrane fraction (pellet from spin 1) was washed once with lysis buffer and ultracentrifuged at 100,000 × $g$, and the pellet was resuspended in lysis buffer with 0.1% SDS and was referred to as membrane fraction.

**Immunoblotting.** A total of 10 µg of WCL was used for immunoblotting or subcellular localization for detection of TNT, EsxF, and EsxE. RNAP was used as a loading control for WCLs and as a lysis control for in vitro secretion assays, GlpX was used as a water-soluble compartment control, and MctB was used as a membrane compartment control for subcellular localization. A total of 50 µg was used to detect EsxF or EsxE due to the weakness of the polyclonal antibodies. LpqH was used as a secreted protein control as it is found in extracellular vesicles of Mtb. Samples were prepared with Laemmli protein loading dye and run on a 10% SDS–PAGE gel and transferred overnight with a PVDF membrane. Blots were blocked with Odyssey blocking buffer in TBST (LI-COR Biosciences) with two washes with TBST between detection steps. Primary antibody detection was performed using rabbit polyclonal anti-TNT (1:1000), rabbit polyclonal anti-EsxF (1:50), rabbit polyclonal anti-EsxE (1:50), mouse monoclonal anti-RNAP (1:1000, BioLegend), mouse monoclonal anti-LpqH (1:1000, BEI Resources), rabbit polyclonal anti-GlpX (1:1000, BEI Resources), and anti-MctB (1:1000). Secondary detection was performed with donkey anti-rabbit IgG-IRDye 680RD or goat anti-mouse IgG-IRDye 800CW, which were obtained from LI-COR biosciences.

**Cross-linking experiments**. Mtb strains were grown to late log phase and were washed three times with PBS + tyloxapol, and 3 mM of DCC was added to the culture and incubated f or 30 min at room temperature. The reaction was quenched using 10 mM glycine and incubated for 15 min. Cells were then prepared for immunoblotting, as described above.

**Fluorescence microscopy of mycobacteria**. Mtb cultures were grown to log phase in AMTB medium (OD = 0.6) and a 1 ml aliquot was stained overnight with 100 μg/ml DMN-trehalose at 37 °C on an end-over-end rotor to label Mtb outer membranes and live cells (final OD = ~1.5). The next day, the stained cells were spun at $3000 \times g$ and washed once with PBS. The cells were then fixed with 4% paraformaldehyde in PBS for 20 min at room temperature. The cells were then spun at $3000 \times g$ and washed twice with PBS. The cells were then blocked with 5% normal donkey serum (Thermo Fisher Scientific) for 30 min, and then incubated with 1:50 α-TNT antibody for 1 h at room temperature. Cells were then washed three times with PBS to remove unbound antibody and then incubated with 1:100 donkey anti-rabbit IgG (H + L)–Alexafluor-594 conjugate for 1 h to detect bound TNT antibodies. Cells were then washed three times with PBS to remove unbound secondary antibody. Bacteria were imaged on agarose pads using a 100× oil immersion lens on a Zeiss Axiovert 200 inverted fluorescence microscope (Carl Zeiss). Fluorescence acquisition settings for exposure and gain were normalized to the Δ*esxFE-cpnT-ifT* strain (baseline, nonspecific antibody binding), and all strains were imaged using the same settings. For quantification, the acquired red channel image was split in imageJ to remove spectral overlap from the blue and green channels. The resulting image was then analyzed with the find maxima tool using a threshold of 15 to count individual punctae. The find maxima tool was then used on the corresponding brightfield image to count Mtb cells. At least 500 cells were counted per strain per experiment over five fields of view from two independent experiments for a total of at least 1000 bacterial cells for the full analysis. Data were then represented as the mean percentage of TNT+ cells per field of view over two independent experiments. The same experiment was repeated using α-esxF or α-esxE antibodies, using the same protocol except that Δ*esxFE-cpnT-ifT::+psmyc-cpnT-ifT* strain was used to normalize the background to account for induction of other Esx proteins, as a consequence of *cpnT* expression.

**Flow cytometry of bacteria**. Mycobacterial strains were grown under identical conditions for immunofluorescent microscopy with the following changes. The cells were not labeled with DMN-trehalose, and were blocked with 5% normal goat serum. The secondary antibody used for detection was goat anti-rabbit IgG-FITC conjugated (Thermo Fisher Scientific). Flow cytometry was performed on a BD Accuri 6 cytometer. The buffer was run as a control followed by unstained bacteria with acquisition limits of 1,000,000 events or 150 μl of sample. All fluorescently labeled strains were then collected and represented as histograms. The Δ*esxFE-cpnT-ifT* strain was used to determine the background levels of antibody binding. The raw plots are shown in the Source data file.

**THP-1 cell culture and infection**. THP-1 monocytes (ATCC TIB-202) were cultured at 37 °C with 5% CO₂ in Roswell Park Memorial Institute (RPMI) complete medium (2 mM L-glutamine, 25 mM HEPES, 10% fetal bovine serum, 1% non-essential amino acids, and 1% antibiotic/antimycolate solution) at minimal density of $1 \times 10^5$ cells/ml to a maximum of $8 \times 10^5$ ml before passaging. Monocytes were seeded on sterile coverslips in 24-well plates at a density of $4 \times 10^5$ cells per well with 100 nM PMA overnight. The indicated Mtb mc²6206 strains were grown in 10 ml AMTB medium to log phase (OD = 1.0) and 1 ml of culture was labeled with DMN-trehalose[72,73] overnight at 37 °C on an end-over-end rotor protected from light. The following morning adherence of THP-1s was verified by phase-contrast microscopy and the medium was washed three times with warm RPMI Infection medium (RPMI + 10% normal human serum (Thermo Fisher Scientific) and without antibiotics). The Mtb strains were then spun at $3000 \times g$ and washed two times with infection medium and sonicated in a water bath for 1 m to disperse clumps. Cells were then diluted to the appropriate density for multiplicity of infection of 10 Mtb/macrophage (OD = 1 = $3 \times 10^8$ Mtb/ml) in RPMI infection medium and were sonicated in a water bath for an additional 1 m prior to infection. The Mtb-infected THP-1s were incubated for 4 h at 37 °C with 5% CO₂. The media was then removed and cells were washed with warm RPMI complete medium + 20 μg/ml gentamycin and incubated for 2 h to kill extracellular bacteria. The cells were then washed three times with RPMI complete (without antibiotics) and incubated for 48 h prior to processing.

**Fluorescence microscopy of THP-1 cells infected with Mtb**. Mtb-infected THP-1s were harvested after 48 h and the medium was aspirated. The cells were then fixed with 4% paraformaldehyde (Electron Microscopy Sciences) in PBS (cell culture grade, Corning) for 20 min at room temperature. The cells were washed three times with PBS and quenched with 0.1 M glycine, followed by three washes with PBS. Cells were then permeabilized with 20 μg/ml digitonin in PBS (to permeabilize the plasma membrane and detect antigens in the macrophage cytosol only) for 5 m with gentle agitation and then washed three times with PBS. Cells were then blocked with 5% donkey serum in PBS for 30 m on an orbital shaker. Rabbit polyclonal TNT, EsxF, and EsxE antibodies were used at a concentration of

(1:100) in PBS and cells were incubated for 1 h on an orbital shaker. Cells were washed two times with PBS and incubated with donkey anti-rabbit Alexafluor-594 at 1:500 concentration for 1 h on an orbital shaker. Cells were then washed three times with PBS and then mounted with ProLong Glass Antifade Mounting medium on cover slides protected from light overnight. Slides were analyzed on a Zeiss Axiovert 200. For image acquisition, all samples were normalized to the negative control: Δ*esxFE-cpnT-ifT* and the exposure and gain settings were lowered until background was barely visible in the Alexafluor-594 channel. All subsequent images were taken under the same acquisition settings. For analysis: infected cells were scored as being Alexafluor-594+ (for TNT, EsxF, or EsxE in the respective experiment) compared to the negative controls. At least 100 cells per strain over four independent experiments were analyzed and quantified as a percentage of Alexafluor-594+ cells over the total cells analyzed.

**Protein expression and purification**. Gene expression plasmids were transformed into *E. coli* BL21(DE3) cells for protein production. A starter culture from a single colony was incubated for 8 h in 25 ml lysogeny broth supplemented with 1% glucose and 100 μg/ml carbenicillin. Seed stocks were made in 15% glycerol and saved at −80 °C until needed. Protein expression was performed via auto-induction in ZYP5052 medium[74]. A single seed stock was used to inoculate 1 l of ZYP5052 with 100 μg/ml carbenicillin, and the culture was grown at a 1/5th volume/flask aeration rate for 24 h at 37 °C. The culture was then harvested by centrifugation at $3000 \times g$ and the pellet was resuspended in lysis buffer (50 mM sodium phosphate, 300 mM NaCl, pH 7.4, 10% glycerol, 1 mM PMSF, 1 μg benzonase (NOVAGEN) per 10 ml of buffer, and one Roche Complete™ protease inhibitor cocktail tablet per 30 ml of sample). Samples were sonicated using a 1/4 in. tip probe at 50 watts for 30 s on/off for a total of 30 min on ice. Samples were then spun at $50,000 \times g$ to pellet debris and membrane fractions. The soluble fraction was then run over 15 ml NEB amylose resin pre-equilibrated with lysis buffer. The column was washed with 50 ml of 25 mM sodium phosphate, 150 mM NaCl, pH 7.0 (cold spring harbor protocols recipe), and then eluted in 50 ml of the same buffer with 20 mM maltose monohydrate (Sigma). The average yield of MBP fusion protein per liter was ~150 mg. The MBP fusion was then cleaved by TEV protease at 1 μg TEV/100 μg fusion protein without shaking at 4 °C overnight. Imidazole was added to the samples at a final concentration of 30 mM from a stock solution and the mixture was run over a Ni-NTA nickel resin (Thermo; 10 ml) to bind His₆-MBP, and His₆-TEV, and uncleaved fusion protein. The IMAC column was washed once with the same buffer containing 30 mM imidazole and pooled with the flowthrough, since we observed that some EsxE-EsxF protein sticks to nickel resins due to its net negative charge. The sample was then subjected to anion exchange using a HiTrap QFF column on a Bio-Rad DuoFlow FPLC with 25 mM sodium phosphate, 150 mM NaCl pH 7.0 as buffer A and 25 mM sodium phosphate, 2 M NaCl, pH 7.0 as buffer B. We used 150 mM NaCl as the input and washing buffer as it is highly effective in removing contaminating MBP, which elutes at 150 mM NaCl (from the NEB amylose resin manual), whereas EsxE-EsxF eluted at ~400–500 mM salt concentration. The sample was then concentrated using an Amicon ultrafiltration tube with a 3 kDa cutoff at $3000 \times g$ using a swinging bucket centrifuge. This sample was referred to as anion exchange fraction. For gel filtration and isolation of different oligomeric species, the sample was separated on a Superdex 200 (GE Life Sciences) using 25 mM sodium phosphate 150 mM NaCl pH 6.5. Samples for EM or lipid bilayer analysis were used immediately after purification or saved at −20 °C in 50% glycerol for subsequent analysis.

**Planar lipid membrane experiments**. Lipid bilayer analysis was performed on a custom lipid bilayer apparatus[31]. All membrane painting events were performed with a filter rise time of 30 ms, constant voltage at −10 mV, with an amplification factor of $1 \times 10^6$ using a Keithley 428 Current Amplifier. All data were recorded with an amplification factor at $1 \times 10^9$. Teflon cuvettes were impregnated with 10 mg/ml of 100% 1,2-diphytanoyl-sn-glycero-3-phosphocholine (DiphPC) lipids (Avanti Polar Lipids) in chloroform for 1 h at room temperature. The cuvette was filled with 5 ml of bilayer buffer (25 mM sodium phosphate, 1 M KCl at pH 6.5, or whatever pH was required for the experiment) on each side to bathe the electrodes and a membrane was painted with 1 mg/ml DiphPC in n-decane solution, and allowed to run for at least 1 h to test stability. All lipids were kept in solution at −20 °C. For channel activity experiments, 1 μg of EsxE-EsxF anion exchange fraction was added to the cis and trans side of the cuvette. For all experiments with pH or with EsxE-EsxF mutants, we performed the experiment under identical conditions for ease of comparison. Optimal channel activity was defined as at least 100 channels over ten membranes from at least two independent preparations of protein. To test the effect of phosphate on channel activity the bilayer buffer was changed to 10 mM HEPES, 1 M KCl pH 6.5. Raw data were analyzed using IGOR Pro 5.03 (WaveMetrics) using a custom algorithm. The data were further analyzed in Graphpad Prism to generate figures and perform statistical analysis. For the pH-shift assay, the experiment was performed at pH 6.5 until EsxE-EsxF pores inserted until saturation. HCl was then added to the cuvette until the final pH was 5.5 and data were recorded for an hour post addition.

**Electron microscopy**. Negatively stained samples were prepared using glow-discharged (15 s at 40 mA) continuous carbon or continuous carbon on lacey film on copper grids. A total of 3 µl of protein sample was placed on the grid for 1 m followed by blotting with Whatman paper, two times washes with milli-Q water, and then staining 2× with 1% uranyl formate (Electron Microscopy Sciences), first for 15 s, then for 1 min. Grids were air-dried and saved at room temperature until imaged. Ideal protein concentration was determined empirically, with a final concentration of 10–50 nM as the ideal range, assuming 150 kDa as the molecular weight for all calculations. Electron microscopy was performed using an FEI Tecnai F20 microscope operating at 200 kV, equipped with a Gatan K2 or K3 direct electron detector operating in electron counting mode. Images were collected using Digital Micrograph at a magnification of 40,500× resulting in a pixel size of 1.23 Å in the specimen, at defocus values between 0.5 and 1.96 µm, and an exposure time of 1 s. At least 50 micrographs were collected per protein sample.

**Electron microscopy image processing**. Data processing was carried out using EMAN2.2 (ref. [75]), using the EMAN2.2 tutorial. Approximately 500 particles were manually chosen across multiple micrographs using a particle size of 100 and box size of 150 × 150 pixels and 50 reference-free 2D class averages were calculated and used as references for particle autopicking of the entire dataset. CTF correction was performed automatically and validated manually using a pixel size of 1.23 Å/pixel. The selected particles were then subjected to five rounds of 2D class averaging.

**Electron microscopy 3D reconstruction and model building**. All image processing and 3D reconstruction was carried out using EMAN2.2. The indicated particle averages in Fig. S8A were used to generate an initial model with C5 symmetry (Fig. S8b). The model with projection images (Fig. S8c) that best matched the reference-free 2D class averages was selected for further refinement with C5 symmetry. The initial model was then refined for two iterations with a target resolution of 25 Å (Fig. S8d). The resolution of the final reconstruction was estimated using a Fourier shell correlation criterion of 0.143 and was calculated to be 17.6 Å (Fig. S8e). The EsxEF homology model was generated in SwissModel[76] (Supplementary Data 1, file 1), the disordered C-terminus of EsxF was deleted, and the resulting structure (Supplementary Data 1, file 2) was then fitted into the refined density map, using rigid-body fitting with the "fit in map" tool in Chimera (Fig. S8F)[77] (Supplementary Data 1, file 3). For Rosetta-based de novo structure generation, the EsxEF homology model was used as the input on the symmetric docking program on the Rosetta Online Server that Includes Everyone (ROSIE; Fig. S8g)[78]. C5 symmetry was selected to generate 1000 de novo docking structures. The lowest free energy models were selected and validated docking using rigid-body fitting in Chimera (Fig. S8h). The model which best represented the experimental density and rigid-body fitting is provided in Supplementary Data 1 (file 4).

**Liposome binding experiments**. DMPC (Avanti Polar Lipids) liposomes were prepared by extrusion with a 100 nm polycarbonate membrane in 25 mM sodium phosphate, 150 mM NaCl pH 6.5. EsxE-EsxF was prepared in the same buffer using an Amicon Ultracentrifugal filter with a 3 kDa MW cutoff. A final solution of 1 mg/ml DMPC liposomes + 50 nM EsxE-EsxF was prepared in the same buffer and was incubated for 30 min at room temperature and 3 µl was prepared for negative stain electron microscopy in 1% uranyl formate.

**Giant unilamellar vesicle membrane deformation assay**. GUVs were prepared with the desired lipids using electroporation in a Nanion Technologies Vesicle Prep Pro system in a buffer containing 25 mM sodium phosphate (pH 6.5) and 150 mM NaCl. The GUVs were then labeled with the lipophilic dye FM-464X (Thermo Fisher Scientific) at 5 µM for 1 h at room temperature. GUV's were then treated with 40 µg of EsxE-EsxF or the indicated WXG variants for 30 min at room temperature. GUV's were then imaged on a coverslip with a slice of 1% agarose on top to prevent movement. The coverslip was then imaged inverted with with oil with a 100× objective on the same microscope described previously. GUVs treated with buffer only were used as a control to determine morphology. At least 25 fields of view were taken per experiment for a total of at least 50 GUVs per experiment. Results are representative of two independent experiments.

**NAD+ glycohydrolase assay**. Mtb strains were grown to log phase in AMTB medium and were spun at 3000 × g and the pellet was lysed in 25 mM sodium phosphate buffer, 150 mM NaCl pH 6.5 + 1 mM PMSF, and one Roche cOmplete protease inhibitor tablet using a bead beater. Half of the sample was then treated at 85 °C for 5 min to denature IFT and then cooled for 5 min on ice (heat-treated sample). A total of 2.5 µg of lysate was then incubated with 100 µM NAD+ in lysis buffer for 30 min at 37 °C and then quenched 1:1 with 10 M sodium hydroxide for 1 h at room temperature protected from light. The fluorescence of the samples was then measured at 460 nm after excitation at 360 nm using a BioTek Synergy plate reader. Recombinant TNT was purified as described previously[79] and was used as a control (100 ng).

**Differential scanning fluorometry**. Differential scanning fluorimetry measures the temperature-dependent fluorescence of proteins, and was used to determine the thermostability of EsxE-EsxF and its variants (Tycho or Prometheus, NanoTemper).

**Figures**. Figures were made in Coreldraw 17 or using Biorender (Fig. 7). Coreldraw 17 is licensed to Michael Niederweis. Biorender account is licensed to Uday Tak.

**Reporting summary**. Further information on research design is available in the Nature Research Reporting Summary linked to this article.

## Data availability

The electron density map for EsxEF is deposited in the electon microscopy data bank (EMDB code: EMD-22727). Rosetta generated PDB files and rigid-body fitting models are provided in Supplementary Data 1 (containing four files). Any additional data is available upon request. Research material described in this study is available upon request. Source data are provided with this paper.

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

## Acknowledgements
We thank Dr. Olga Danilchanka for initial experiments with *esxF* and *esxE*, Ying Wang for help with the EsxE and EsxF antibody production, Dominik Herrmann and Dr. Mikhail Pavlenok for assistance with lipid bilayer experiments, and Cynthia Rodenberg for assistance with electron microscopy experiments. DMN-trehalose for metabolic labeling of mycobacteria was kindly provided by Drs. Mireille Kamariza and Carolyn Bertozzi (Stanford University). We thank Dr. Jim Sun for constructing the Mtb Δ*esxFE-cpnT-ifT* strain and Dr. Avishek Mitra for assistance with the PDIM analysis of this strain. We thank Dr. Moon Nahm and Rob Burton for use of their flow cytometer and help with data interpretation. Electron microscopy data analysis was performed on the UAB CHEAHA computing platform, which is supported by the National Science Foundation grant nos. OAC-1541310, the University of Alabama at Birmingham, and the Alabama Innovation Fund. We thank Dr. Peter Prevelige for help with MS experiments and Drs. Peter Prevelige and Jamil Saad for valuable scientific discussions. This work was supported by the National Institutes of Health grant R01 AI121354 to M.N.

## Author contributions
M.N. conceived the project, U.T. and M.N. designed the experiments; U.T. and T.D. performed research; U.T., T.D., and M.N. wrote and edited the manuscript; M.N. and T.D. directed research.

## Competing interests
The authors declare no competing interests.
