## [Peer Review File · Nature Communications]

REVIEWER COMMENTS

Reviewer #1 (Remarks to the Author):

In this highly relevant, rigorous and well written study, the authors define a novel way used by proteins to cross the mycobacterial membranes- via a transmembrane pore made up of WXG family proteins EsxE and EsxF. They show that these proteins are required for the secretion of CpnT and that they form a transmembrane channel through the mycobacterial OM. This paper is very strong and will clearly move the field of mycobacterial protein secretion forward. We propose editorial changes that we think will make the manuscript clearer.

Concerns:

There are several times throughout the manuscript where the authors, while trying to provide context for a result, make claims that are not directly supported by the data. These statements should be clearly delineated from the conclusions, and perhaps moved to the discussion. For example,

the claim in lines 56-58 that EsxE and EsxF are required for the stability of the CpnT toxin is not supported by the data. While the levels of CpnT protein are reduced in the absence of EsxE and EsxF, the mechanism underlying this observation is unclear- while protein stability is one explanation, the authors have not ruled out that EsxE and EsxF are required for cpnT transcription, transcript stability or translation. Please adjust the claim to indicate that EsxE and EsxF are required to maintain WT levels of CpnT in the mycobacterial cell, or similar. In the discussion, the authors could suggest that it may be protein stability, and may also comment that EsxE and EsxF likely require each other for stability similar to other pairs of Esx proteins including EsxA and EsxB (Renshaw et al, PMID: 11940590 and Stanley et al PMID: 14557536, Okkels et al, PMID: 15060053 and others).

Similarly the claim in Line 70, regarding the idea that CpnT probably gets degraded to prevent self intoxication in the cytosol is not supported by the data- this is one possibility as to why the authors are observing reduced CpnT levels and perhaps why reduced CpnT levels may be significant biologically. This sentence should be very clearly separated from the conclusion, and likely belongs in the discussion.

If there are other instances please likewise move these to the discussion.

B. I would caution you on your conclusion that because mutation of the YXXXDE motif did not impact secretion of EsxE and EsxF, they must be secreted independently of ESX systems.

The YXXXE/D was shown to be required for the secretion of some proteins by the Wilbert Bitter group; it is not sufficient for secretion. The presence of the YXXXE/D motif is not strictly correlated with secreted mycobacterial proteins. The fact that mutations in the YXXXED motif does not impact the secretion of EsxF or EsxE, or CpnT, does not rule out that an ESX system translocates EsxE and EsxF.

Targeting requires interaction between the substrate and components of the ESX systems. The C-terminal 7 AA of EsxB (LSSQMGF), is an ESX targeting signal because it mediates interactions with EccCb1 in the ESX-1 system. Single point mutations in this motif prevents interaction of EsxB with the EccCb1 ESX-1 component and prevents secretion through the ESX-1 system. This has been demonstrated genetically by Champion et al in 2006 (PMID: 16973880), and confirmed using crystallography by Rosenberg et al in 2015 (PMID: 25865481) and Wang et al in 2020 (PMID: 31758528). Co-structures of EsxB and EccCb1 revealed that the only residues from EsxB that interact directly with EccCb1 reside within the 7 terminal amino acids. There appears to be conservation between the 7 terminal residues between EsxB and EsxF, in particular EsxFL97, and the residues that follow, which may mediate direct interaction with an EccC protein and be

required for secretion.

Please address the above points in your discussion, and throughout the manuscript. I think they would strengthen your conclusions, and provide a nice path forward for the research. Please address the possibility that the C-terminal AA of EsxF could be the targeting signal, since it shares residues with EsxB that are in fact required for targeting and are in the secretion signal. Please cite the suggested references. These papers may also explain why your mutations in the YXXXE/D motif do not impact secretion.

Patricia A Champion

Reviewer #2 (Remarks to the Author):

The manuscript by Tak et al. addresses an intriguing question of the TNT toxin secretion by *M. tuberculosis*. They show that small WXG100 proteins – EsxE-EsxF – could form a range of multimeric structures in vitro and demonstrate a channel properties. From these results, the authors conclude that EsxE-EsxF could be self-sufficient to mediate the secretion of TNT. This is an interesting study that presents some novel ideas about protein secretion in *M. tuberculosis*. However, some data appear to be overinterpreted, and alternative explanations could be considered for some of the findings. Some of the data could benefit from additional statistical analysis. The major and minor points are outlined below.

Major points:

Line 11: There is no data presented in the manuscript regarding EsxE-EsxF location in inner and/or outer membrane of *M. tuberculosis*.

Line 67: The authors show the TNT levels are decreased on the cell surface. However, as shown in Fig. 1b, the total protein level of CpnT is also greatly reduced. Therefore the lack of the CpnT surface localization could be explained by the lower protein levels beyond limit of detection in fluorescence experiments.

Line 72: Self-intoxication seems an unlikely reason for TNT degradation, as the level of immunity protein is unaffected by the absence of EsxE-EsxF (Fig. 1b).

Line 100: It is stated that the level of CpnT is unaffected by Y86A mutation in esxF, however, the CpnT band in Fig. 2b appears weaker than in the WT esxF background. Some explanation should be provided in addition to statistical analysis of the immunoblot data.

Line 131: It is difficult to reconcile the data on the membrane localization and the surface localization of EsxE-EsxF. Both proteins are present in the membranes (Fig. 3a), form an obligatory dimer or higher order oligomer, yet only EsxF is surface-localized?

Line 159: A comparison Figure with the hydrophobicity surfaces of EsxE-EsxF vs. VacA, YaxAB and cytolysin A would be really helpful to illustrate similarities.

Line 167: Discrepancy between the native PAGE and size-exclusion chromatography data should be explained - EsxE-EsxF behaves mostly as a dimer on the native PAGE, but an oligomer on SEC?

Line 168: The 'proteins' should elute at 10 kDa under the denaturing conditions – presumably the heterodimer of EsxE-EsxF would be disrupted at 6M GuHCl. The peak appears to be eluting earlier than 44 kDa standard, not at 20 kDa as stated in the text.

Data availability: EM maps and models should be deposited to the EMDB and the PDB.

The connection between the ESX secretion systems and CpnT secretion should be addressed. It is quite possible that EsxE-EsxF represent a functional module of one of the ESX systems. The authors stated that EsxE-EsxF could function as specific chaperone, which would explain the data presented. The Mtb strains lacking ESX-1, ESX-3 and ESX-5 systems have been described and could be tested for secretion of CpnT.

Minor points:

Line 33: Reference formatting.

Line 53: Correct to 'Table S2'?

Lines 123-124: Reference for Figs 1,2 appears to be misplaced.

Fig. S2 is not referenced in the text.

Line 158: Fig. S3C appears to be missing, or mis-referenced.

Reviewer #3 (Remarks to the Author):

In this manuscript, Tak and coworkers investigate the function of two small proteins belonging to the Esx family of type VII secretion - associated proteins of Mycobacterium tuberculosis and show conclusive evidence that they form a membrane channel, which is essential for secretion of tuberculosis necrotizing toxin (TNT).

The authors present very convincing data that the two proteins EsxE and EsxF are required for translocation of the toxin to the Mtb cell surface, and further into the host cell. They also show that the conserved WxG motifs are essential for this function. The most exciting part of the manuscript - in my opinion- is the data that the EsxE-EsxF complex forms membrane spanning pores, and that for the interaction with membranes the WxG motifs are required.

The paper is well written and the data convincing and presented very clearly.

Thus I only have minor suggestions for the improvement of the manuscript:

The authors describe the WxG motif, but do not cite the original paper by Pallen Trends Microbiol. 2002 . Please change.

The EsxE and EsxF proteins are encoded outside of any bigger ESX cluster, as stated by Tekaiia et al, Tuber Lung Dis. 1999; or Bitter et al; PLoS Pathog. 2009.

in line 385, the authors discuss that the previous pore forming activity of ESAT-6 was an artifact due to a detergent used in the preparation of heterologously produced ESAT-6 and cited the paper by Conrad et al.. However, this sentence should be written with more care as the same paper also showed that ESX-1 was necessary for membrane lysis. The authors could also refer to a very recent paper by Augenstreich et al., where the effect of the detergent is confirmed, but where a preparation of native ESAT-6 confirmed lytic activity of ESAT-6 (without impact of detergents)

Please see Augenstreich et al.

(<https://www.frontiersin.org/articles/10.3389/fcimb.2020.00420/full>). The authors further report about filaments formed by EsxE-EsxF. It would be interesting to compare their data in the discussion to the data presented on EspC (an Esx-associated protein with the same small size as Esx proteins) Lou et al., Mol Microbiol. 2017.

Rebuttal – NCOMMS-20-30312

Manuscript title: Pore-forming Esx proteins mediate toxin secretion in *Mycobacterium tuberculosis*

Authors: Uday Tak, Terje Dokland, and Michael Niederweis

Reviewer 1:

1. The claim in line 56-68 that EsxE and EsxF are required for the stability of the CpnT toxin is not supported by the data. While the levels of CpnT protein are reduced in the absence of EsxE and EsxF, the mechanism underlying this observation is unclear- while protein stability is one explanation, the authors have not ruled out that EsxE and EsxF are required for cpnT transcription, transcript stability or translation. Please adjust the claim to indicate that EsxE and EsxF are required to maintain WT levels of CpnT in the mycobacterial cell, or similar. In the discussion, the authors could suggest that it may be protein stability, and may also comment that EsxE and EsxF likely require each other for stability similar to other pairs of Esx proteins including EsxA and EsxB (Renshaw et al, PMID: 11940590 and Stanley et al PMID: 14557536, Okkels et al, PMID: 15060053 and others).

We agree with Dr. Champion that we did not directly rule out transcriptional regulation of CpnT levels by EsxE and EsxF. Therefore, we only conclude in the results part that: "Collectively, these results show that both EsxE and EsxF are required to maintain CpnT protein levels in Mtb" (lines 60-61).

In the discussion we further elaborate:

"The fact that the amount of IFT protein, encoded by the last gene in the *cpnT* operon, is not affected by the absence of the *esxF-esxE* genes, indicates that transcription and, due to the likely translational coupling of the overlapping *cpnT-ift* genes {Levin-Karp, 2013 #148}, also translation of the resulting mRNA are not changed. Thus, in the absence of EsxE and EsxF, CpnT stays in the Mtb cytosol and is either bound by the immunity factor IFT and/or degraded to prevent self-poisoning as shown previously⁹ (lines 452-457).

2. Similarly the claim in Line 70, regarding the idea that CpnT probably gets degraded to prevent self-intoxication in the cytosol is not supported by the data- this is one possibility as to why the authors are observing reduced CpnT levels and perhaps why reduced CpnT levels may be significant biologically. This sentence should be very clearly separated from the conclusion, and likely belongs in the discussion.

This comment was clearly a speculation based on our previous findings that TNT is toxic when expressed in Mtb or in *E. coli* without co-expression of the antitoxin (Sun et al, *Nature Structural and Molecular Biology*, 2015). In the revised manuscript, we have removed this comment from the results and discuss this possibility as described above (lines 452-457).

We also added a question mark behind "Toxin degradation" in our model (Fig. 7) to account for the fact that we did not formally show degradation of CpnT protein.

3. I would caution you on your conclusion that because mutation of the YXXXDE motif did not impact secretion of EsxE and EsxF, they must be secreted independently of ESX systems. The YXXXE/D was shown to be required for the secretion of some proteins by the

Wilbert Bitter group; it is not sufficient for secretion. The presence of the YXXXE/D motif is not strictly correlated with secreted mycobacterial proteins. The fact that mutations in the YXXXED motif does not impact the secretion of EsxF or EsxE, or CpnT, does not rule out that an ESX system translocates EsxE and EsxF.

We agree with this statement. Indeed, we never concluded that EsxE and EsxF are secreted independently of ESX systems. We wrote that “Considering the association of many small Esx proteins with the T7S systems and the presence of a T7SS signal sequence YXXXD/E in EsxF (Fig. 2A), it is possible that the EsxE EsxF heterodimer interacts with an ESX system...” (ll. 349-351, original manuscript). We have now deleted our previous conclusion that “it is unclear whether the EsxE-EsxF-dependent toxin secretion pathway depends on one of the existing ESX machineries in Mtb.” Instead, we have added to our discussion the results mentioned by Dr. Champion below (see answer to comment 4). (Lines 364-372).

4. Targeting requires interaction between the substrate and components of the ESX systems. The C-terminal 7 AA of EsxB (LSSQMGF), is an ESX targeting signal because it mediates interactions with EccCb1 in the ESX-1 system. Single point mutations in this motif prevents interaction of EsxB with the EccCb ESX-1 component and prevents secretion through the ESX-1 system. This has been demonstrated genetically by Champion et al in 2006 (PMID: 16973880), and confirmed using crystallography by Rosenberg et al in 2015 (PMID: 25865481) and Wang et al in 2020 (PMID: 31758528). Co-structures of EsxB and EccCb1 revealed that the only residues from EsxB that interact directly with EccCb1 reside within the 7 terminal amino acids. There appears to be conservation between the 7 terminal residues between EsxB and EsxF, in particular EsxFL97, and the residues that follow, which may mediate direct interaction with an EccC protein and be required for secretion.

We thank Dr. Champion for pointing out that there are similarities in the C-termini of EsxF and EsxB. We discuss the possibility that this sequence might be a play a role in targeting EsxF to an ESX system (Lines 368-372).

Reviewer 2

1. Line 11: There is no data presented in the manuscript regarding EsxE-EsxF location in the inner and/or outer membrane of *M. tuberculosis*

In subcellular fractionation experiments we show that app. half of the EsxF and EsxE proteins are associated with the membrane fraction (Fig. 3A). We further show that EsxF is accessible on the cell surface (Fig. 3B). This two-step method has been established by us as evidence for outer membrane localization in several publications (*Nat Struct Mol Biol* **22**, 672; *Mol Microbiol* **97**, 881; *MBio* **8**, e01720) and is now increasingly accepted in the TB field (see for example *Science* **367**, 1147). In addition, we show that the EsxEF complex interacts with membranes (Fig. 4I) and forms pores in lipid bilayers (Fig. 4F) demonstrating that the EsxEF proteins form an integral membrane-spanning complex. These experiments conclusively demonstrate that at least a part of the EsxEF complex is localized in the outer membrane.

However, we agree that the expression “channel-tunnel” is speculative and have changed this term to “outer membrane channel”.

2. Line 67: The authors show the TNT levels are decreased on the cell surface. However, as shown in Fig. 1b, the total protein level of CpnT is also greatly reduced. Therefore the lack of the CpnT surface localization could be explained by the lower protein levels beyond limit of detection in fluorescence experiments.

It is possible that the absence of EsxE and EsxF could reduce CpnT levels beyond the limit of detection in the microscopy experiments. However, the EsxE_{W38A} mutation also reduced CpnT levels (Fig. 2B), but TNT can be detected on the cell surface albeit at much lower levels than wild type (Fig. 2F-G). This suggests that the wt EsxE-EsxF complex is involved in translocating CpnT to the cell surface.

3. Line 72: Self-intoxication seems an unlikely reason for TNT degradation, as the level of immunity protein is unaffected by the absence of EsxE-EsxF (Fig. 1b).

See same comment to question 2 of reviewer 1.

This comment was clearly a speculation based on our previous findings that TNT is toxic when expressed in Mtb or in *E. coli* without co-expression of the antitoxin (Sun et al, *Nature Structural and Molecular Biology*, 2015). In the revised manuscript, we have removed this comment from the results and discuss this possibility as described above (lines 452-457).

However, we want to emphasize that it is possible that not all TNT is immediately bound by IFT and, thus, degradation of cytoplasmic CpnT might be an alternative pathway of reducing its toxicity.

4. Line 100: It is stated that the level of CpnT is unaffected by Y86A mutation in esxF, however, the CpnT band in Fig. 2b appears weaker than in the WT esxF background. Some explanation should be provided in addition to statistical analysis of the immunoblot data.

It is obvious that the CpnT band in the Mtb strain producing EsxF_{Y86A} is weaker in the western blot than the CpnT band in the parent strain. However, the same is true for the RNA polymerase (RNAP) band, which was used as a loading control. The CpnT band intensities are identical when normalized to those of the RNA polymerase in the same sample. Thus,

the CpnT levels are the same in both strains. The obvious lower CpnT band intensity in the sample from the Mtb strain producing EsxF_{Y86A} is the result of a loading error. In order to clarify this, we included the quantitative densitometric analysis in Fig. 2C. This is not a statistical analysis.

5. Line 131: It is difficult to reconcile the data on the membrane localization and the surface localization of EsxE-EsxF. Both proteins are present in the membranes (Fig. 3a), form an obligatory dimer or higher order oligomer, yet only EsxF is surface-localized?

The surface exposure of different subunits completely depends on the structure of the complex. It is easily conceivable that only EsxF forms a surface exposed domain or that the epitope(s) of EsxE, which are recognized by our anti-serum, are buried within the protein core or shielded by the lipid bilayer. These hypotheses are supported by our observation that the EsxE-W38A-EsxF mutant forms particles with a different conformation (Fig. 6), which are slightly detectable on the cell surface of Mtb. We have expanded on this for clarity (lines 133-137).

6. Line 159: A comparison Figure with the hydrophobicity surfaces of EsxE-EsxF vs. VacA, YaxAB and cytolysin A would be really helpful to illustrate similarities.

We created a figure to compare the hydrophobicity of the surfaces of EsxE-EsxF, VacA, YaxAB and cytolysin A and added it to the supplement (Fig. S9).

7. Line 167: Discrepancy between the native PAGE and size-exclusion chromatography data should be explained - EsxE-EsxF behaves mostly as a dimer on the native PAGE, but an oligomer on SEC?

This is a valid point. We observed that EsxE/EsxF quickly oligomerizes in solution and eventually assembles to long filaments. This is shown in Figure S4. Although gel filtration gives rather accurate estimates of the molecular weight for globular proteins, which do not interact with column material, helical proteins and protein filaments often do not elute at the correct size due to their divergent shapes and/or interactions with the resin. This is now mentioned in the text (lines 173-176).

8. Line 168: The 'proteins' should elute at 10 kDa under the denaturing conditions – presumably the heterodimer of EsxE-EsxF would be disrupted at 6M GuHCl. The peak appears to be eluting earlier than 44 kDa standard, not at 20 kDa as stated in the text.

In a native gel the EsxE/EsxF complex runs with an apparent molecular weight under 20 kDa indicating that the complex is denatured (Fig. S4h). The reviewer is correct that the complex elutes from the gel filtration column similar to the 44 kDa standard. This error was corrected in the text (line 172). Similarly, we have clarified the discrepancies between the apparent molecular weights observed using different techniques and the migration of proteins based on hydrodynamic radius during gel filtration (Lines 173-176).

9. Data availability: EM maps and models should be deposited to the EMDB and the PDB.

We uploaded the EM map, half maps, FSC curves, and masks to the EMDB (EMD-22727). We have requested that the map be released following publication. This is now mentioned in the Data Availability section (lines 764-765) and in Fig. 6A (line 923).

The PDB files from rigid body fitting and Rosetta-generated symmetric docking are not experimentally derived structures. Rigid body fitting was done using the experimental EM map as a guide. The Rosetta-generated model was fit into the EM density map, but was generated in the absence of any guiding density. The *de novo* structure strongly resembles our experimental density thus validating the general shape and features of the particle. However, the resolution of our negative stain reconstruction is too low to identify subunit orientation or any atomic details. For this reason, we prefer to not post pdf files of these structures. However, the coordinates of these models are available upon request.

10. The connection between the ESX secretion systems and CpnT secretion should be addressed. It is quite possible that EsxE-EsxF represent a functional module of one of the ESX systems. The authors stated that EsxE-EsxF could function as specific chaperone, which would explain the data presented. The Mtb strains lacking ESX-1, ESX-3 and ESX-5 systems have been described and could be tested for secretion of CpnT.

In several instances we mentioned the possible connection between EsxE-EsxF and ESX systems. In the revised version, we have rewritten the paragraph discussing the possible connection between EsxE-EsxF and ESX systems (364-372).

We already showed that ESX-1 is not required for CpnT secretion (Sun et al, *Nature Structural and Molecular Biology* 2015). However, examining whether and how EsxE-EsxF interact with an ESX system to secrete TNT is beyond the scope of this manuscript.

Minor points:

11. Line 33: Reference formatting

Fixed.

12. Line 53: Correct to 'Table S2'?

Fixed

13. Lines 123-124: Reference for Figs 1,2 appears to be misplaced

The reference to Figures 1 and 2 was made to emphasize that we observed CpnT in the fraction of water-soluble proteins in other experiments as well. We now write "(see also Figs. 1, 2)" to avoid confusion.

14. Fig S2 is not referenced in the text

Figure S2 is referenced at line 251.

15. Line 158 Fig. S3C appears to be missing, or mis-referenced

We have adjusted this to include Figures S3A-S3D which includes MemSAT analysis, homology modeling, and delineation of the hydrophobic domains in the structural model.

Reviewer 3:

1. The authors describe the WxG motif, but do not cite the original paper by Pallen Trends Microbio. 2002. Please change.

We cited the Pallen reference in our original manuscript (ref 57, l. 445). Now, we added the reference to the Pallen paper also in the introduction, when we first mention the WXG motif of Esx proteins (line 31).

2. The EsxE and EsxF proteins are encoded outside of any bigger ESX cluster, as stated by Tekaiia et al, Tuber Lung Dis. 1999; or Bitter et al; PLoS Pathog. 2009.

Figure 1A shows *esxFE* as part of the *cpnT* operon. We now added the reference to the Bitter et al. paper to the introduction (line 34) in which *esxF-esxE* were originally described as “orphaned *esx* genes”.

3. in line 385, the authors discuss the that the previous pore forming activity of ESAT-6 was an artifact due to a detergent used in the preparation of heterologously produced ESAT-6 and cited the paper by Conrad et al.. However, this sentence should be written with more care as the same paper also showed that ESX-1 was necessary for membrane lysis. The authors could also refer to a very recent paper by Augenstreich et al., where the effect of the detergent is confirmed, but where a preparation of native ESAT-6 confirmed lytic activity of ESAT-6 (without impact of detergents) Please see Augenstreich et al (<https://www.frontiersin.org/articles/10.3389/fcimb.2020.00420/full>).

We have rephrased this statement and focused only on the pore-forming activity of Esat-6 (EsxA). Specifically, we write (lines 407-412):

However, a recent report concluded that “ESAT-6 does not function as a pore-forming protein or possess intrinsic membrane lytic activity under physiological conditions” and that the previously observed pore-forming activity of EsxA was an artifact of a detergent used during purification{Conrad, 2017 #16}. While this conundrum is not yet solved for EsxA, in this study the EsxE-EsxF complex was purified in the absence of detergents and showed channel activity in lipid bilayer experiments without detergents (Fig. 4F,G, 5C,D).

Since we did not discuss the lytic activity of ESAT6, we did not add the Augenstreich et al. paper.

4. The authors further report about filaments formed by EsxE-EsxF. It would be interesting to compare their data in the discussion to the data presented on EspC (an Esx-associated protein with the same small size as Esx proteins) Lou et al., Mol Microbiol. 2017.

We mention now in the discussion the filaments formed by EspC. We also include the reference to a paper by Grant Jensen, in which his group shows filaments in the periplasm of *M. smegmatis* using cryo electron tomography (lines 378-381).

Comment:

We also edited the manuscript to correct typos and improve language.

All changes are marked in a pdf file named “EsxEF_Manuscript_v107_Changes.pdf”.

REVIEWERS' COMMENTS

Reviewer #2 (Remarks to the Author):

The addressed all questions.

Small correction:

"738 (Figure S8E). The EsxEF homology model was generated in SwissModel76, the disordered c-terminus". Correct to "C-terminus".